# MMBench: Is Your Multi-modal Model an All-around Player?

## Abstract

Large vision-language models have recently achieved remarkable progress, exhibiting great multimodal perception and reasoning abilities. However, how to effectively evaluate these large vision-language models remains a major obstacle, hindering future development in this domain. Traditional benchmarks like VQAv2 or COCO Caption provide quantitative performance measurements but suffer from a lack of fine-grained ability assessment and non-robust evaluation metrics. Recent subjective benchmarks, such as OwlEval, offer comprehensive evaluations of a model's abilities by incorporating human labor, but they are not scalable and display significant bias. In response to these challenges, we propose MMBench, a new benchmark for assessing multi-modal capabilities of VLMs. MMBench methodically develops a comprehensive evaluation pipeline, primarily comprised of two key features: 1. MMBench is a meticulously curated dataset that surpasses existing similar benchmarks in terms of the number and the variety of evaluation questions and abilities; 2. MMBench introduces a rigorous CircularEval strategy and incorporates the use of ChatGPT to convert free-form predictions into predefined choices, thereby facilitating a fair and robust evaluation despite of VLMs' different instruction following capabilities. MMBench is a systematically designed objective benchmark for robustly evaluating the various abilities of vision-language models. We hope MMBench will assist the research community in better evaluating their models and facilitate future progress in this area.

## 1 Introduction

Recently, notable progress has been achieved within the realm of large language models (LLMs). For instance, the latest large language models, such as OpenAI's ChatGPT and GPT-4 (OpenAI, 2023), have demonstrated remarkable reasoning capabilities that are comparable to, and in some cases, even surpass human capabilities. Drawing inspiration from these promising advancements in LLMs, large vision-language models (LVLMs) have also experienced a revolutionary transformation. Notable works, such as MiniGPT-4 (Zhu et al., 2023), Otter (Li et al., 2023b;a), and LLaVA (Liu et al., 2023b), have demonstrated enhanced capabilities in terms of image content recognition and reasoning within the domain of vision-language models, demonstrating superior capabilities compared to early works. Nevertheless, the majority of current studies tend to emphasize showcasing qualitative examples, rather than undertaking comprehensive quantitative experiments to thoroughly assess their model performance. The lack of quantitative assessment poses a considerable challenge for comparing various models. Recent studies have mainly explored two approaches to conduct quantitative evaluations. The first approach involves utilizing existing public datasets (Goyal et al., 2017; Chen et al., 2015) for objective evaluation. Alternatively, some studies employ human annotators (Ye et al., 2023; Xu et al., 2023) to perform subjective evaluation. However, it is worth noting that both approaches exhibit some inherent limitations.

A multitude of public datasets, such as VQAv2 (Goyal et al., 2017), COCO Caption (Chen et al., 2015), GQA (Hudson & Manning, 2019), OK-VQA (Marino et al., 2019), have long served as valuable resources for the quantitative evaluation of vision-language models. These datasets offer **objective** metrics, including accuracy, BLEU, CIDEr, *etc*. However, when employed to evaluate more advanced LVLMs, these benchmarks encounter the following challenges:

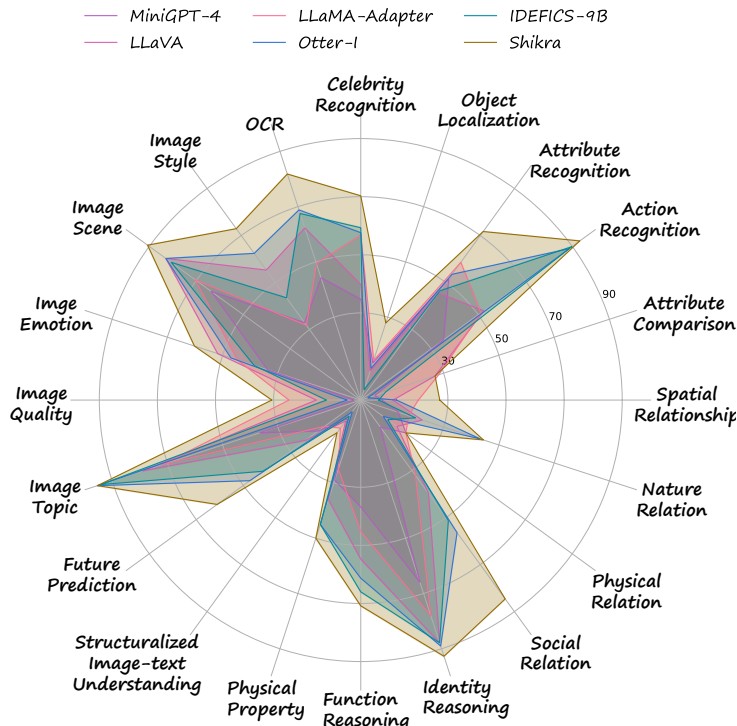

Figure 1: **Results of six representative large vision-language models across the 20 ability dimensions defined in MMBench.** For more comprehensive evaluation results on additional models, please refer to Table 3 and Table 8, as well as the appendix.

**(1).** Existing evaluation metrics mostly require an exact match between the prediction and the reference target, leading to potential limitations. For instance, in the Visual Question Answering (VQA) task, even if the prediction is "bicycle" while the reference answer is "bike", the existing metric would assign a negative score to the prediction, despite its correctness. Consequently, this issue results in a considerable number of false-negative samples.

**(2).** Current public datasets predominantly focus on evaluating a model's performance on some specific tasks, offering limited insights into the fine-grained capabilities of these models. Thus, they provide insufficient feedback regarding potential directions for future improvements.

Given the aforementioned challenges, recent studies, such as mPLUG-Owl (Ye et al., 2023) and LVLM-eHub (Xu et al., 2023) propose human-involved **subjective** evaluation strategies, aiming to address existing methods' limitations by incorporating human judgment and perception in the evaluation process. mPLUG-Owl comprises 82 artificially constructed open-ended questions related to 50 images sourced from existing datasets. After predictions are generated by both mPLUG-Owl and another vision-language (VL) model, human annotators will assess the quality of these predictions. Similarly, inspired by FastChat (Zheng et al., 2023), LVLM-eHub develops an online platform where two models are prompted to answer a question related to an image. A participant then compares the answers provided by the two models. These subjective evaluation strategies offer several advantages, consists of **accurate matching** (humans can accurately match a prediction to the target, even if presented in different words) and **comprehensive assessment** (humans tend to compare two predictions based on multiple aspects). The final score is calculated as the average score across different abilities, enabling a comprehensive evaluation of various model capabilities.

While subjective evaluation allows for a more comprehensive assessment of a model's abilities, it also introduces new challenges that need to be addressed. Firstly, human evaluations are inherently biased. Consequently, it becomes challenging to reproduce the results presented in a work with a different group of annotators. Also, existing subjective evaluation strategies face scalability issues. Employing annotators for model evaluation after each experiment is an expensive endeavor. Moreover, evaluation datasets of small sizes can result in statistical instability. To ensure a robust evaluation, collecting more data becomes necessary, which in turn demands a significant amount of human labor.

In light of the challenges faced by conventional objective and subjective benchmarks, we propose **MMBench**, a systematically designed objective evaluation benchmark to robustly evaluate different abilities of large vision-language models. Currently, MMBench contains approximately 3000 single-choice questions covering 20 different ability dimensions, such as object localization and social reasoning, for evaluating vision-language models. Each ability dimension includes more than 75 questions, enabling a balanced and comprehensive evaluation of various abilities. The ability dimensions are not static and will expand as we continue to work on them. Since some existing VLMs have poor instruction-following capability and cannot directly output choice labels (A, B, C, *etc.*) for MMBench questions, the evaluation based on exact matching may not yield fair and reasonable conclusions. In order to reduce the number of false-negative samples during answer matching, we employ ChatGPT to match a model's prediction to one of the choices in a multi-choice question and then output the label for the matched choice. We conduct a comparison between ChatGPT-based choice matching and human evaluations, and discovered that ChatGPT can accurately match human assessments in 87% of cases. Further investigation on the misaligned cases shows that ChatGPT yields more correct matchings compared to human annotators, demonstrating its good alignment and robustness as an evaluator. To make the evaluation more robust, we propose a novel evaluation strategy, named **CircularEval** (details in Sec. 3.1). We comprehensively evaluate 18 well-known vision-language models on MMBench and report their performance on different ability dimensions. The performance ranking offers a direct comparison between various models and provides valuable feedback for future optimization. In summary, our main contributions are three-fold:

- **Systematically-constructed Dataset**: To thoroughly evaluate the capacity of a VLM, we have carefully curated a dataset, comprised of a total of 2,948 meticulously selected questions, which collectively cover a diverse spectrum of 20 fine-grained skills.
- **Robust Evaluation**: We introduce a novel circular evaluation strategy (CircularEval) to improve the robustness of our evaluation process. After that, ChatGPT is employed to match model's prediction with given choices, which can successfully extract choices even from predictions of a VLM with poor instruction-following capability.
- **Analysis and Observations**: We perform a comprehensive evaluation of a series of well-known vision-language models using MMBench, and the evaluation results can provide insights to the research community for future improvement.

## 2 MMBENCH

There exist two unique characteristics that differentiate MMBench from existing benchmarks for multi-modality understanding: i) MMBench adopts problems from various sources to evaluate diversified abilities in a hierarchical taxonomy; ii) MMBench applies a robust, LLM-based evaluation strategy, which can well handle the free-form outputs of multi-modality models and yield trustworthy evaluation results with affordable cost. In this section, we focus on the first characteristic and organize the subsequent content as follows: In Sec. 2.1, we present the hierarchical ability taxonomy of MMBench and discuss the design philosophy behind. In Sec. 2.2, we briefly introduce how we collect the MMBench questions, and provide some statistics of MMBench.

### 2.1 THE HIERACHICAL ABILITY TAXONOMY OF MMBENCH

Human possess remarkable perception and reasoning capabilities, allowing them to understand and interact with the world. These abilities have been crucial in human evolution and serve as a foundation for complex cognitive processes. Perception refers to gathering information from sensory inputs, while reasoning involves drawing conclusions based on this information. Together, they form the basis of most tasks in the real world, including recognizing objects, solving problems, and making decisions (Oaksford & Chater, 2007; Fodor, 1983). In pursuit of genuine general artificial intelligence, vision-language models are also expected to exhibit strong perception and reasoning abilities. Therefore, we incorporate **Perception** and **Reasoning** as our top-level ability dimensions in our ability taxonomy, referred to as **L-1** ability dimension. After that, the **L-2** and **L-3** ability dimensions, belonging to **Perception** and **Reasoning**, are also derived. More details about these ability dimensions are shown in Figure 2. Detailed definitions of all L-3 abilities are presented in Appendix F.

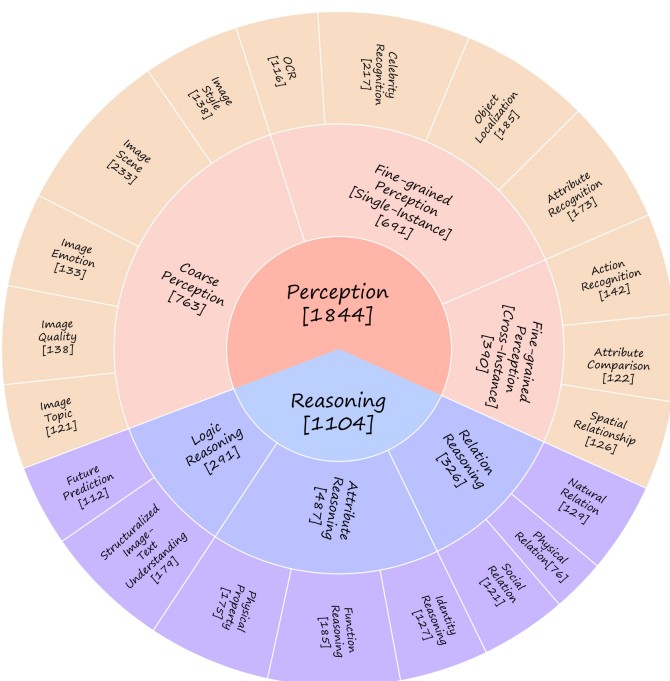

Figure 2: **Overview of existing ability dimensions in MMBench.** Currently, MMBench incorporates three levels of ability dimensions (from L-1 to L-3), which encompass 20 distinct leaf abilities. Note that the demonstrated ability taxonomy is an initial version. Further adjustments can be made to improve the taxonomy and make it more comprehensive.

## 2.2 DATA COLLECTION AND STATISTICS

In the current version of MMBench , we collect vision-language QAs in the format of multiple-choice problems for each L-3 ability. A multiple choice problem $P_i$ corresponds to a quadruple $(Q_i, C_i, I_i, A_i)$. $Q_i$ denotes the question, $C_i$ represents a set with $n$ ($2 \leq n \leq 4$) choices $c_1, c_2, ..., c_n$, $I_i$ corresponds to the image associated to the question, and $A_i$ is the correct answer. In Figures 7 to 12, we visualize data samples corresponding to each **L-3** ability. Our data — which include images, choices, and questions — are manually collected from multiple sources by a group of volunteers from colleges. Before starting the collection process, we equip our volunteers with the necessary training. The training includes: i) Detailed definitions for each ability dimension, from L1 to L3. ii) Potential sources for image, question, and choice collection. If no existing questions or choices are available, the volunteers are trained on how to create them. A comprehensive list of these sources can be found in Table 7. iii) We compile a set of 10 to 50 multiple-choice problems for each L-3 ability. These problems serve as exemplars, demonstrating the specific types of problems associated with the evaluation of each respective ability. Using these examples as references, the annotators can expand the collection of multiple-choice problems for each L-3 ability. This ensures that the collected problems remain relevant and suitable for assessing the targeted abilities. iv) The process of saving the collected or created data samples, including the necessary content and meta-information, is also covered in the training. It is noteworthy that some data samples originate from public datasets such as COCO-Caption (Chen et al., 2015), which has been used by several public vision-language models in pre-training. Regardless, evaluation on MMBench can still be considered as out-domain evaluation (Dai et al., 2023) for two primary reasons: Firstly, our data is gathered from the validation sets of these public datasets, not their training sets. Secondly, data samples procured from these public datasets constitute less than 10% of all MMBench data samples.

**Data Statistics.** In the present study, we have gathered a total of 2,948 data samples spanning across 20 distinct **L-3** abilities. We depict the problem counts of all the 3 levels of abilities in Figure 2. To ensure a balanced and comprehensive evaluation for each ability, we try to maintain an even distribution among problems associated with different abilities during data collection.

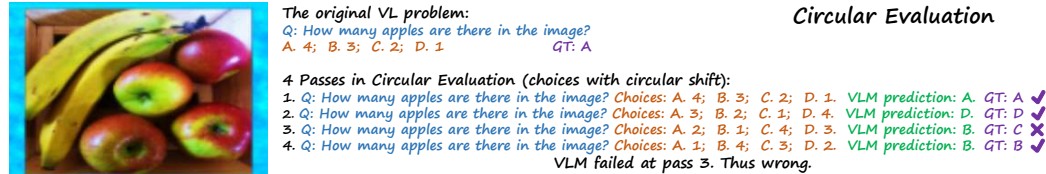

Figure 3: **A demonstration of the Circular Evaluation strategy.** In Circular Evaluation, a problem is tested multiple times with circular shifted choices and the VLM needs to succeed in all testing passes. In this example, the VLM failed in pass 3 and thus considered failed the problem.

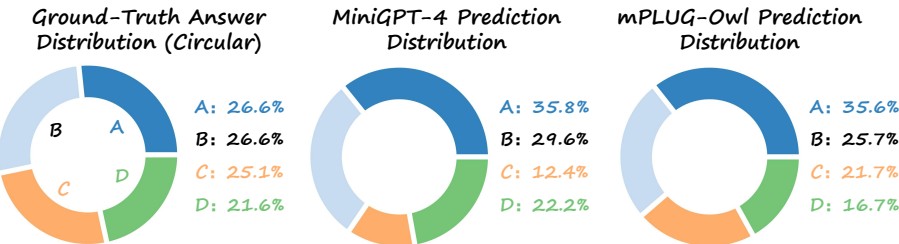

Figure 4: **The choice distribution of ground-truth answers and predictions of MiniGPT-4 and mPLUG-Owl (all *CircularEval* records).** Since there exist questions with only 2 or 3 choices in MMBench , the choice distribution of ground-truth answers is not exactly balanced.

**Data Splits.** We follow the standard practice in previous works (Marino et al., 2019) to split MMBench into `dev` and `test` subsets at a ratio of 4:6. For the `dev` subset, we make all data samples publicly available along with the ground truth answers for all questions. For the `test` subset, only the data samples are released, while the ground truth answers remain confidential. To obtain the test subset evaluation results, one needs to submit the predictions to MMBench evaluation server.

**Data Cleaning.** In preliminary study, we find that it's possible to solve some questions in MMBench solely without resorting to the associated image. In some cases, the question can be solved with external knowledge, while in other cases the quality of distractors is not good. To this end, we create a subset of MMBench that excludes all questions that can be correctly answered by GPT-4, and evaluate the VLMs on the subset. Detailed results can be found in Appendix A.2.

## 3 EVALUATION STRATEGY

In MMBench we propose a new evaluation strategy that yields robust evaluation results with an affordable cost. At a strategic level, we adopt the **Circular Evaluation** strategy, which feeds a question to a VLM multiple times (with different prompts) and checks if the VLM succeeds in solving the question in all attempts. To deal with the free-form VLMs' outputs, we propose to utilize ChatGPT as a helper for choice extraction. We conduct extensive experiments to study the ChatGPT-involved evaluation procedure. The results well support the effectiveness of ChatGPT as a choice extractor. Without specifications, we use **gpt-3.5-turbo-0613** as the choice extractor in all of the following experiments.

### 3.1 THE CIRCULAR EVALUATION STRATEGY

MMBench incorporates a diverse range of problems aimed at assessing the multifaceted capabilities of vision-language models (VLMs). These problems are presented as multiple-choice questions. The formulation poses an evaluation challenge: random guessing can lead to ∼25% Top-1 accuracy for 4-choice questions, potentially reducing the discernible performance differences between various VLMs. Besides, we noticed that VLMs may perfer to predict a certain choice among all given choices (Figure 4), which further amplify the bias in evaluation. To this end, we introduce a more robust evaluation strategy termed **Circular Evaluation** (or **CircularEval**). Under this setting, each question is fed to a VLM $N$ times ($N$ equals to the choice number). Each time circular shifting is applied to the choices and the answer to generate a new prompt for VLMs (example in Figure 3). A VLM is considered successful in solving a question only if it correctly predicts the answer in all rotational

passes. CircularEval doesn't necessarily requires $N\times$ API calls. By definition, if the VLM made a wrong prediction in one pass, we can directly drop the following passes and say the VLM fails to answer this question. CircularEval can achieve a good trade-off between the robustness and the evaluation cost.

## 3.2 CHATGPT-INVOLVED CHOICE EXTRACTION

In our initial attempts to solve the MMBench questions, we observed that the instruction-following capabilities of various VLMs is limited. Though problems are presented as clear multiple-choice questions with well-formatted options, many VLMs still output the answers in free-form text (*e.g.*, model's output can be ` The correct answer is [choice "A" content] `, but not ` A `). Extracting choices from free-form predictions is easy for human beings, but difficult with rule-based matching. Thus we design a universal evaluation strategy for all VLMs with different instruction-following capabilities:

**Step 1. Matching Prediction.** Extract choices from VLM predictions with exact matching. For each choice, we try to match its choice label (the uppercase letter, with optional prefixes or suffixes[1]) or its choice content with content output by a VLM . If there exists one and only one choice that is matched with the VLM's output, we successfully extract the model's choice.

**Step 2. Matching ChatGPT's output.** If step 1 fails, we then try to extract the choice with ChatGPT. We provide GPT with the question, options, and model prediction, and then, we request ChatGPT to align the prediction with one of the given options, and subsequently produce the label of the corresponding option. The specific prompt querying ChatGPT can be found in Appendix D.

**Step 3. Fallback: Random Assignment.** If step 2 can still not extract the choice, we label the prediction with a random choice among all valid choices and 'X'. Additionally, a comment message will be added to denote that ChatGPT fails to parse the model prediction. This step is never encountered is our preliminary feasibility analysis (Sec. 3.3), but we still add it for pipeline integrity.

For each input, we compare the model's label prediction (after GPT's similarity readout) with the actual ground truth label. If the prediction matches the label, the test sample is considered correct.

## 3.3 CHATGPT AS THE JUDGE: A FEASIBILITY ANALYSIS

We first conduct pilot experiments to study the effectiveness of ChatGPT as the judge. To keep the setting simple, for MMBench , we sample a subset with ∼1000 samples and use the vanilla single-pass evaluation strategy to evaluate 8 selected VLMs.

Table 1: **Success rate of each step in our choice extraction.**

| Model Name | Step-1 | Step-2 |
|---|---|---|
| LLaMA-Adapter (Gao et al., 2023) | 1.0% | 100.0% |
| OpenFlamingo (Alayrac et al., 2022) | 98.6% | 100.0% |
| VisualGLM (Du et al., 2022) | 14.9% | 100.0% |
| MiniGPT-4 (Zhu et al., 2023) | 71.6% | 100.0% |
| LLaVA (Liu et al., 2023b) | 9.9% | 100.0% |
| Otter-I (Li et al., 2023b) | 100.0% | 100.0% |
| InstructBLIP (Dai et al., 2023) | 91.2% | 100.0% |
| mPLUG-Owl (Ye et al., 2023) | 42.6% | 100.0% |

**Instruction following capabilities of different VLMs vary a lot.** ChatGPT-involved choice extraction plays a vital role in MMBench evaluation, especially for VLMs with poor instruction following capabilities. In Table 1, we demonstrate the success rate of step 1 and step 2 of our evaluation strategy. Step 1 success rate (matching choices with VLM predictions) is directly related to the VLM's instruction-following capability. Table 1 shows that the step-1 success rates of different VLMs vary a lot, covering a wide range from 1.0% to 100.0%.

With ChatGPT choice extractor equipped, the step-2 success rates of all VLMs reach 100%, which enables a fair comparison of different VLMs on MMBench. Another point worth noting is, the instruction following capability and the overall multi-modality modeling capability is not necessarily correlated. OpenFlamingo (Alayrac et al., 2022) demonstrates top instruction following capability among all VLMs, while also achieving one of the worst performance on MMBench (Table 3).

**Human *vs* ChatGPT: alignment in choice extraction.**

For VLM predictions that cannot be parsed with exact matching, we adopt ChatGPT as the choice extractor. To validate its efficacy, we sample a subset of MMBench , which contains 103 questions and 824 (103 × 8) question-answer pairs. We keep only the QA pairs that can not be parsed by the

---

[1]As an example, for 'C', we try to match ` "C", "C.", "C)", "C,", "C)." `, *etc.* . Since 'A' may serve has an article in a sentence, we skip ` "A" ` during matching.

Table 2: **CircularEval** *vs* **VanillaEval.** We compare CircularEval and VanillaEval on MMBench `dev` split and present the overall Top-1 accuracy of all VLMs.

| Eval \VLM | OpenFlamingo | LLaMA-Adapter | MiniGPT-4 | MMGPT | InstructBLIP | VisualGLM | LLaVA | Qwen-VL |
|---|---|---|---|---|---|---|---|---|
| **VanillaEval** | 34.6% | 62.6% | 56.9% | 49.1% | 61.7% | 61.0% | 62.8% | 60.6% |
| **CircularEval** | 4.6% | 41.2% | 32.0% | 15.2% | 35.5% | 38.6% | 44.5% | 38.2% |
| Δ | **-30.0%** | **-21.4%** | **-24.9%** | **-33.9%** | **-26.2%** | **-22.4%** | **-18.3%** | **-22.4%** |
| Eval \VLM | OpenFlamingo v2 | mPLUG-Owl | MiniGPT-4-13B | Otter-I | InstructBLIP-13B | PandaGPT | Shikra | Qwen-VL-Chat |
| **VanillaEval** | 40.0% | 65.5% | 61.3% | 69.2% | 64.8% | 55.2% | 70.4% | 75.6% |
| **CircularEval** | 6.7% | 48.1% | 42.5% | 51.6% | 44.4% | 33.9% | 59.4% | 60.6% |
| Δ | **-33.3%** | **-17.4%** | **-18.8%** | **-17.6%** | **-20.4%** | **-21.3%** | **-11.0%** | **-15.0%** |

evaluation step 1, which yield 376 data samples. With the help of 6 volunteers, we perform manual choice extraction to these data samples[2].

In Figure 5, we report the alignment rate (extracted choices are exactly the same) between ChatGPT and Human. Specifically, ChatGPT (GPT-3.5) achieves 87.0% alignment rate, while the more powerful GPT-4 achieves a slightly better 87.2%. We further conduct an ablation study to learn the effect of using various LLMs as the choice extractor. GPT-4 and ChatGPT take the lead among all LLMs. Claude achieves a very close alignment rate (86.4%) compared to ChatGPT. Existing open-source LLMs adapted from LLaMA (Gao et al., 2023) and GLM (Du et al., 2022) achieves poor performance on the choice matching task. Further scaling the architecture (*e.g.* from Vicuna-7B to Vicuna-33B) only leads to limited improvements. We adopt ChatGPT as the choice extractor in our evaluation for a good performance-cost trade-off.

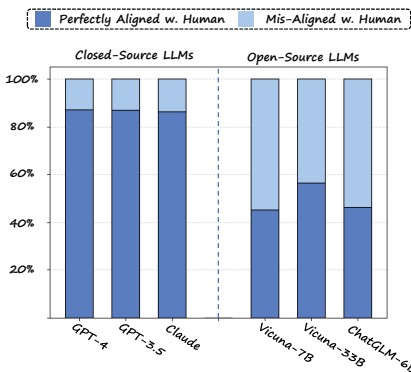

Figure 5: **The alignment rate between human and different LLMs in choice extraction.**

## 4 EVALUATION RESULTS

### 4.1 VLM INFERENCE SETTING

Currently, we adopt the traditional zero-shot setting for VLM inference, primarily due to the limited compatibility of existing VLMs with few-shot evaluation settings. However, we have noticed the great potential of few-shot evaluation protocols in LLMs (Huang et al., 2023). In future work, we specifically plan to construct a subset of data samples designated for few-shot evaluation. We anticipate that few-shot evaluation will evolve into a standard assessment strategy, akin to the approach employed in LLMs.

### 4.2 MAIN RESULTS

We select 18 different multi-modality models (containing some variants of a same model family) and benchmark them on MMBench. The models we have selected cover a broad spectrum of strategies and architectures, effectively illustrating the current state-of-the-art in multimodal understanding. To facilitate a fair comparison, we mainly examine the "light" versions of all multimodal models — those with a total amount of parameters below 10B — when multiple variants exist. For further reference, we also evaluate larger variants (*e.g.* 13B, 80B) of some selected models, and report their performance. Please refer to Table 6 for detailed information regarding the architecture and the total parameters of these models.

Before delving deeper into concrete evaluation results, we first compare our **CircularEval** (infer a question multiple passes, consistency as a must) with **VanillaEval** (infer a question only once). In Table 2, we present the results with two evaluation strategies on MMBench `dev` split. For most VLMs,

---
[2]The human annotations will be released.

Table 3: **CircularEval results on MMBench `test` set (L-2 abilities).** Abbreviations adopted: LR for Logical Reasoning; AR for Attribute Reasoning; RR for Relation Reasoning; FP-C for Fine-grained Perception (Cross Instance); FP-S for Fine-grained Perception (Single Instance); CP for Coarse Perception. The dash line separate models with the parameter size $\leq$ **10B** or $>$ **10B**.

| VLM | Overall | CP | FP-S | FP-C | AR | LR | RR |
|------|---------|------|------|------|------|------|------|
| **OpenFlamingo (Alayrac et al., 2022)** | 4.5% | 1.5% | 2.5% | 1.6% | 12.5% | 9.8% | 3.3% |
| **OpenFlamingo v2 (Alayrac et al., 2022)** | 5.7% | 4.1% | 5.3% | 0.8% | 12.8% | 11.6% | 1.4% |
| **MMGPT (Gong et al., 2023)** | 16.0% | 18.4% | 17.8% | 5.3% | 24.0% | 1.2% | 20.9% |
| **MiniGPT-4 (Zhu et al., 2023)** | 29.4% | 36.6% | 35.2% | 16.6% | 42.4% | 12.1% | 14.2% |
| **Qwen-VL (Bai et al., 2023)** | 32.2% | 36.4% | 32.9% | 27.9% | 43.1% | 9.8% | 30.3% |
| **VisualGLM (Du et al., 2022)** | 33.6% | 41.3% | 35.7% | 18.2% | 49.0% | 11.6% | 28.0% |
| **InstructBLIP (Dai et al., 2023)** | 35.4% | 43.7% | 34.4% | 25.1% | 49.7% | 22.0% | 22.3% |
| **LLaMA-Adapter (Zhang et al., 2023b)** | 39.6% | 50.7% | 44.7% | 33.6% | 47.6% | 13.3% | 23.2% |
| **LLaVA (Liu et al., 2023b)** | 42.2% | 59.3% | 37.9% | 30.8% | 57.6% | 15.0% | 27.0% |
| **IDEFICS-9B (Laurençon et al., 2023)** | 45.5% | 56.3% | 43.2% | 36.8% | 64.9% | 21.4% | 29.4% |
| **mPLUG-Owl (Ye et al., 2023)** | 46.4% | 60.4% | 53.0% | 25.1% | 55.6% | 16.2% | 40.3% |
| **Otter-I (Li et al., 2023b;a)** | 48.5% | 60.8% | 46.7% | 36.4% | 63.5% | 22.5% | 39.8% |
| **Shikra (Chen et al., 2023)** | 60.4% | 71.9% | 61.8% | **50.6%** | 69.8% | 33.5% | **53.1%** |
| **Qwen-VL-Chat (Bai et al., 2023)** | **61.8%** | **72.8%** | **66.3%** | 46.2% | **74.3%** | **40.5%** | 47.9% |
| **PandaGPT (Su et al., 2023)** | 30.7% | 47.8% | 20.1% | 20.6% | 41.7% | 15.6% | 22.3% |
| **MiniGPT-4-13B (Zhu et al., 2023)** | 42.5% | 50.7% | 49.7% | 19.8% | 62.8% | 17.3% | 30.3% |
| **InstructBLIP-13B (Dai et al., 2023)** | 43.4% | 54.2% | 45.7% | 24.3% | 59.7% | 17.3% | 36.5% |
| **IDEFICS-80B (Laurençon et al., 2023)** | 54.8% | 62.1% | 56.0% | 48.6% | 68.1% | 29.5% | 46.4% |

switching from VanillaEval to CircularEval leads to a significant drop in model accuracy. In general, comparisons under CircularEval can reveal a more significant performance gap between different VLMs. As an example, LLaVA outperforms InstructBLIP by 1.1% top-1 accuracy under VanillaEval, while a much larger performance gap (9.0% top-1 accuracy) is observed under CircularEval. For some specific cases (MiniGPT-4 *vs.* PandaGPT, *e.g.*), the conclusions can be even differenct with two evaluation strategies. In following experiments, We adopt **CircularEval** as our default evaluation strategy, which is a more reasonable and well-defined evaluation paradigm.

We exhaustively evaluate all VLMs on all existing leaf abilities of MMBench. In Table 3, we report the models' overall performance and the performance in six **L-2** abilities, namely Coarse Perception (**CP**), Fine-grained Perception (single-instance, **FP-S**; cross-instance, **FP-C**), Attribute Reasoning (**AR**), Logic Reasoning (**LR**) and Relation Reasoning (**RR**). on the **test** split (**dev** split results in Table 8). These results offer valuable insights into the individual strengths and limitations of each model in different aspects of multi-modality understanding.

As demonstrated in Table 3, Shikra and Qwen-VL-Chat, with localization information integrated in multi-modal learning, yields superior results and significantly outperforms other models across nearly all L-2 abilities. IDEFICS-80B (with LLaMA-65B adopted as the language model) achieves the third best performance, demonstrates strong visual-language modeling capability. After that, three models (Otter-I, mPLUG-Owl, IDEFICS-9B) are roughly at the same level of overall performance, but with strengths in different L2 abilities. Among all VLMs, OpenFlamingo (both versions) and MMGPT demonstrate lower overall performance compared to the other models. We obtain two major findings from the evaluation results: 1. VLMs without supervised finetuning significantly lag behind their finetuned counterparts: OpenFlamingo *vs.* Otter-I, Qwen-VL *vs.* Qwen-VL-Chat, *etc.*; 2. It is apparent that model scaling enhances performance metrics. This is evident as MiniGPT-4-13B outperforms MiniGPT-4 by an impressive 13.1%, and InstructBLIP-13B outperforms its predecessor, InstructBLIP, by a notable 8.0%.

The assessment on MMBench reveals that each multi-modality model exhibits unique strengths and weaknesses across different levels of abilities. This observation highlights the importance of carefully selecting and fine-tuning multi-modality models based on the specific requirements and objectives of a given task. Moreover, the identified limitations in some abilities suggest potential directions for further research and development in multi-modality AI systems.

For a more in-depth understanding, we provide a comprehensive analysis of the L3 abilities in Tables 9 to 14, allowing readers to examine the very details of MMBench and gain deeper insights into the performance disparities among the evaluated models.

## 4.3 ANALYSIS

With the comprehensive evaluation, we observe some interesting facts, which is expected to provide insights for future optimization.

**Existing VLMs have limited instruction-following capabilities.** For the sake of efficient evaluation, we guide each model to output only the label for each option, for instance, A, B, C, or D. However, we observe that these models often generate a full sentence corresponding to one option or a sentence semantically akin to one of the options. This tendency is the primary reason for employing ChatGPT for choice extraction. To improve the usability of multi-modality models to empower diversified applications, pursuing stronger instruction-following ability can be a significant direction.

**The overall performance of existing VLMs is still limited.** The strict **CircularEval** strategy reveals that the overall performance of existing VLMs is not satisfying. In experiments, only IDEFICS-80B, Shikra, and Qwen-VL-Chat succeeded to reach 50% Top-1 accuracy on MMBench `test` for multiple choice questions with at most 4 choices. Potential reasons are two fold: 1. Current VLMs are not robust enough to produce the same prediction with slightly different prompts (performance under **CircularEval** is much worse than the performance under **VanillaEval**, see Table 2). 2. The capabilities of current VLMs are still quite limited and can be further improved. We hope that MMBench and our evaluation strategy can serve as important guideline for future development, iteration and optimization of VLMs in the future.

**Cross-Instance Understanding and Logic Reasoning are extremely difficult.** An examination of our evaluation results reveals that cross-instance understanding—specifically relation reasoning (RR) and cross-instance fine-grained perception (FP-C)—poses a significant challenge for existing Visual Language Models (VLMs). The average accuracy for cross-instance fine-grained perception across all models is 26.0% on the MMBench `test`, significantly lower than that of single-instance fine-grained perception (FP-S, 38.3%). A similar disparity can be observed between relation reasoning (RR) and attribute reasoning (AR) as evidenced in Tables 3 and 8. Furthermore, when compared to other L-2 abilities, the logical reasoning (LR) capability of existing models appears strikingly weak, with an average accuracy of only 17.8%. The results indicate that improving the cross-instance understanding and logic reasoning capabilities of VLMs can be a significant and promising direction.

**The introduction of object localization data is anticipated to enhance model performance.** Among various models, Shikra and Qwen-VL-Chat notably excels, offering significant improvements across almost all L-2 capabilities, particularly in **logical reasoning** and **cross-instance fine-grained perception**. Compared to other opponents, the two models incorporate object localization within their training datasets. The integration of localization data infuses more detailed object-specific information into the models, allowing them to comprehend the dynamic states of objects more effectively. Moreover, it aids in elucidating relationships and interactions between distinct objects. This strategy contributes substantively to the enhancement of the models' capabilities in logical reasoning and cross-instance fine-grained perception.

## 5 CONCLUSION

The inherent limitations of traditional benchmarks (VQAv2, COCO Caption, *etc*.) and subjective benchmarks (mPLUG-Owl, *etc*.) underscore the need for an innovative evaluation paradigm in vision-language understanding. To address this, we introduce MMBench, a multi-modality benchmark that proposes an objective evaluation pipeline of 2,948 multiple-choice questions covering 20 ability dimensions. To produce robust and reliable evaluation results, we introduce a new evaluation strategy named **CircularEval**. The strategy is much stricter than the vanilla 1-pass evaluation and can yield reliable evaluation results with an affordable cost. Additionally, we leverage ChatGPT to match model predictions with target choices, which enables a fair comparison among VLMs with different levels of instruction-following capabilities. Comprehensive studies on both MMBench and public benchmark indicate the feasiblility of using ChatGPT as the judge. We hope MMBench can aid the research community in optimizing their models and inspire future progress.

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

## A    RELATED WORK

### A.1    MULTI-MODAL DATASETS

Large-scale vision-language models have shown promising potential in multi-modality tasks such as complex scene understanding and visual question answering. Though qualitative results so far are encouraging, quantitative evaluation is of great necessity to systematically evaluate and compare the abilities of different VLMs. Recent works evaluated their models on numerous existing public multi-modality datasets. COCO Caption (Chen et al., 2015), Nocaps (Agrawal et al., 2019), and Flickr30k (Young et al., 2014) provide human-generated image captions and the corresponding task is to understand image content and describe it in the form of text. Visual question answering datasets, such as GQA (Hudson & Manning, 2019), OK-VQA (Marino et al., 2019), VQAv2 (Goyal et al., 2017) and Vizwiz (Gurari et al., 2018), contain question-answer pairs related to the given image, used to measure the model's ability on visual perception and reasoning. Some datasets provide more challenging question-answering scenarios by incorporating additional tasks. For example, TextVQA (Singh et al., 2019) proposes questions about text shown in the image, thus involving the OCR task into question-answering. ScienceQA (Lu et al., 2022) focuses on scientific topics, requiring the model to integrate commensense into reasoning. Youcook2 (Zhou et al., 2018) replaces images with video clips, introducing additional temporal information. However, the aforementioned datasets are designed on specific domains, and can only evaluate the model's performance on one or several tasks. Besides, different data formats and evaluation metrics across datasets make it more difficult to comprehensively assess model's capability. Ye et al. (2023) built an instruction evaluation set, OwlEval, consisting of several kinds of visual-related tasks, but in a limited size. Fu et al. (2023) built MME, which is a evaluation dataset containing multi-modality Yes / No questions. However, the exact-matching based evaluation and non-rigorous evaluation setting make it harder to reveal the real performance gap between VLMs. Different from previous works, in this paper, we propose a novel multi-modal benchmark MMBench , which is built based on measuring abilities rather than performing specific tasks, aimed to better evaluating models once and for all.

### A.2    MULTI-MODAL MODELS

Benefit from the success of LLMs *e.g.* GPTs (Radford et al., 2019; Brown et al., 2020; Ouyang et al., 2022), LLaMA (Touvron et al., 2023), and Vicuna (Chiang et al., 2023), multi-modal models also achieve great improvements recently. Flamingo (Alayrac et al., 2022) is one of the early attempts on introducing LLMs into vision-language pretraining. To be conditioned well on visual features, it inserts several gated cross-attention dense blocks between pretrained language encoder layers. OpenFlamingo (Alayrac et al., 2022) provides an open-source version of it. BLIP-2 (Li et al., 2023c) proposes a Querying Transformer (Q-former) to bridge the modality gap between the frozen image encoder and large language encoder. After that, InstructBLIP (Dai et al., 2023) extend BLIP-2 (Li et al., 2023c) with vision-language instruction tuning and achieve better performance. VisualGLM (Du et al., 2022) also adopts Q-former (Li et al., 2023c) to bridge the visual model and language encoder, GLM (Du et al., 2022). LLaVA (Liu et al., 2023b) adopt GPT4 (OpenAI, 2023) only with language inputs to generate instruction-following data for vision-language tuning. Otter (Li et al., 2023b) also constructs an instruction-tuning dataset to improve the ability of instruction-following for OpenFlamingo. MiniGPT-4 (Zhu et al., 2023) believes the ability of GPT4 (OpenAI, 2023) comes from advanced LLMs and proposes to adopt only one project layer to align the visual representation with the language model. Although it is trained in high computational efficiency, it demonstrates some capabilities similar to GPT4 (OpenAI, 2023). mPLUG-Owl (Ye et al., 2023) proposes another learning paradigm by first tuning the visual encoder to summarize and align visual features and then tuning the language model with LoRA (Hu et al., 2021). In this paper, after thoroughly evaluating them on the proposed MMBench and other public datasets, we provide some insights for future multi-modal research.

## B    A VISUAL-ONLY SUBSET OF MMBENCH.

To find out questions in MMBench that can be solved solely based on the text part (without resorting to the image). We try to solve each questions in MMBench given only the question and options with cutting-edge LLMs (Claude-2, GPT-3.5-Turbo (0613 ver.), GPT-4 (0613 ver.)). In the pseudo code

below, we demonstrate how we prompt the LLMs to guide them try their best to make a reasonable guess:

```
prompt_tmpl = """
You are an AI assistant which is designed to answer questions from people.
You will be asked a question and provided several choices,
and you need to choose the best choice to answer the question.
There is an image associated with the problem, but that is not provided to you,
so you need to try your best to hallucinate the image.
Again, you can only choose from the provided choices,
and output a single uppercase letter (A, B, C, D, etc.). \n
Question: <question begins> {} <question ends>; \n
Choices: <choices begin> {} <choices end>. \n
Your answer:
"""
example_question = "How many apples are there in the image?"
option_str = "A. 4\nB. 3\nC. 2\nD. 1"
prompt = prompt_tmpl.format(example_question, option_str)
```

Except the absence of visual information, the testing procedure is exactly the same as testing for VLMs. In the table below, we present the accuracy that those LLM-based approaches can achieve on MMBench-Dev (with CircularEval applied).

| Model | Overall | CP | FP-S | FP-C | AR | LR | RR |
|---|---|---|---|---|---|---|---|
| GPT-4 | 13.23 | 6.41 | 7.84 | 6.99 | 32.16 | 27.96 | 4.34 |
| GPT-3.5 | 11.17 | 2.7 | 9.9 | 4.9 | 22.11 | 25.42 | 10.43 |
| Claude-2 | 10.74 | 5.74 | 6.14 | 2.8 | 27.14 | 22.88 | 4.35 |

As demonstrated in above table, the performance of the text-only GPT-4 model — prompted solely with question text — is notably sub-optimal. This limitation is particularly evident in responding to queries that require 'perception' abilities. Such questions, often intrinsically linked to visual information, include examples like "What mood does this image convey?" or "Who is this person?". In contrast, GPT-4 exhibits a degree of proficiency in 'reasoning' tasks, with a notable strength in logical reasoning (LR). The model's performance in LR is comparable to that of other leading open-source models. The effectiveness in LR could be attributed to the model's ability to leverage common sense in answering questions. Notably, some LR data, sourced from ScienceQA, suggest that a portion of these tasks can be effectively addressed using the model's inherent common sense reasoning (Lu et al., 2022).

Based on the above LLM inference results, we remove the questions that were correctly answered by GPT-4 and re-calculate the accuracy. The results are demonstrated in the table below. Removing those questions has limited impact on the final accuracy, and will not lead to significant changes.

| Model | Dev | Dev w/o. GPT-4 | Test | Test w/o. GPT-4 |
|---|---|---|---|---|
| Qwen-VL-Chat | 60.57 | 59.9 | 61.83 | 60.75 |
| Shikra | 59.36 | 60 | 60.43 | 60.81 |
| IDEFICS-80B | 54.81 | 51.29 | 54.82 | 52.34 |
| IDEFICS-9B | 48.37 | 47.03 | 45.52 | 44.8 |
| mPLUG-Owl | 48.11 | 48.51 | 46.41 | 46.11 |
| LLaVA | 44.5 | 42.77 | 42.21 | 41.74 |
| InstructBLIP-13B | 44.42 | 43.17 | 43.39 | 43.18 |
| MiniGPT-4-13B | 42.53 | 40.5 | 42.54 | 41.62 |
| LLaMA-Adapter | 41.24 | 41.09 | 39.63 | 39.63 |
| VisualGLM | 38.57 | 37.82 | 33.63 | 33.52 |
| Qwen-VL | 38.23 | 36.53 | 32.23 | 31.34 |
| InstructBLIP | 35.48 | 33.66 | 35.37 | 34.08 |
| PandaGPT | 33.93 | 32.77 | 30.72 | 29.1 |
| MiniGPT-4 | 32.04 | 30.69 | 29.43 | 28.04 |
| MMGPT | 15.21 | 13.66 | 15.98 | 15.26 |
| OpenFlamingo v2 | 6.7 | 5.45 | 5.72 | 4.61 |
| OpenFlamingo | 4.64 | 3.37 | 4.54 | 3.55 |

## C    COMPARISON WITH CONTEMPORARY WORKS

MME (Fu et al., 2023) is another significant contemporary work alongside MMBench. Both benchmarks focus on evaluating various capabilities of a vision-language model, while they are different in the following aspects:

**(1). Benchmark Scale & Data Balance.**    MME includes **1187 images** covering **14 leaf ability** categories, each associates with two different questions. Meanwhile, the scale of MMBench is larger, which includes **2948 images** covering **20 leaf ability** categories, divided into a development set and a confidential testing set. Besides, MMBench is **more balanced** across ability categories. MME primarily focuses on perception tasks: **89.0%** questions in MME are for perception tasks, and only **11.0%** questions are for cognition tasks. Besides, **4 leaf abilities** have only **20 image samples**. In MMBench, **62.6%** questions are for perception tasks while **37.4%** are for reasoning ones. Except one leaf ability that contains only **76** images, all other leaf abilities include more than **110** image samples. The larger scale and improved data balance can lead to more stable evaluation results.

**(2). Answer Matching Strategy.**    MME continues to utilize **exact matching** to extract answers from VLMs' predictions, which necessitates the model to produce precise 'yes' or 'no' responses. This approach, however, tends to generate a number of false negative samples.

**(3). Evaluation Metric.**    Metrics adopted by MME (accuracy & accuracy +) enable random guessing to achieve a substantial amount of score (**37.5%**), and also makes it difficult to reveal the performance gap between VLMs. Besides, it appears not reasonable that certain models yield scores on the MME that are lower than those of random guesses (especially for perception tasks). The table below provides some data points of this issue. Notably, MMBench doesn't suffer from the problem. Examples: i) XComposer outperforms Qwen-VL-Chat by **14.2%** Top-1 accuracy on MMBench-Dev, which is a substantial gap. However, in MME, the gap is merely 70 in total score (**2.5%** of the total score). ii) In MME, VisualGLM-6B and MiniGPT-4 demonstrates poorer performance compared to the random baseline, which doesn't sounds reasonable. However, in MMBench-Dev, the two approaches significantly outperforms the random baseline (**<0.5%** Top-1 accuracy).

| Method | MME-Perception | MME-Cognition | MME-Total | MMBench-Dev |
|---|---|---|---|---|
| GPT4-V (OpenAI, 2023) | 1409 | 517 | 1926 | 75.1% |
| XComposer (Zhang et al., 2023a) | 1528 | 391 | 1919 | 74.8% |
| Qwen-VL-Chat (Bai et al., 2023) | 1488 | 361 | 1849 | 60.6% |
| mPLUG-Owl2 (Ye et al., 2023) | 1450 | 313 | 1763 | 66.5% |
| VisualGLM-6B (Du et al., 2022) | 705 | 182 | 887 | 38.1% |
| MiniGPT-4 (Zhu et al., 2023) | 582 | 144 | 726 | 24.3% |
| Random | 750 | 375 | 1225 | <0.5% |

## D    CHOICE EXTRACTION

**ChatGPT-based choice extraction.**    To utilize ChatGPT as the choice extractor, we query it with the our manually created template including the question, options and the corresponding VLM's prediction. We prompt ChatGPT with two hand-crafted examples to improve its instruction-following capability.

```
gpt_query_template = (
    "You are an AI assistant to help me matching an answer with several options of a multiple choice question. "
    "You are provided with a question, several options, and an answer, "
    "and you need to find which option is most similar to the answer. "
    "If the meaning of all options are significantly different from the answer, output X. "\
    "Your should output a single uppercase character in A, B, C, D (if they are valid options), and X. \n"
    "Example 1: \n"
    "Question: What is the main object in image?\nOptions: A. teddy bear B. rabbit C. cat D. dog\n"
    "Answer: a cute teddy bear\nYour output: A\n"
    "Example 2: \n"
    "Question: What is the main object in image?\nOptions: A. teddy bear B. rabbit C. cat D. dog\n"
    "Answer: Spider\nYour output: X\n"
    "Example 3: \n"
    f"Question: {question}?\nOptions: {options}\nAnswer: {prediction}\nYour output: ")
```

We then get the prediction's option (e.g. A) from GPT's response. For most questions, GPT-3.5 is capable of returning a single character (e.g., A, B, C) as the answer.

With the above prompt and the extraction strategy introduced in Sec. 3.2, we successfully readout all VLMs' predictions for questions in MMBench. In Figure 5, we list the alignment rates with human of different LLMs when adopted as the choice extractor and show the advantage of ChatGPT. We further investigate the cause of this misalignment since the choice extraction result of ChatGPT is not entirely aligned with that of humans.

## D.1 MIS-ALIGNED CASES ANALYSIS.

Due to the newly introduced pseudo choice 'X', sometimes humans and LLMs can make different decisions when doing choice extraction due to different matching thresholds. For example, agent $\mathcal{A}$ may match the prediction $\mathbf{P}$ with a given choice $\mathbf{C}$ since $\mathbf{C}$ is the most similar choice to $\mathbf{P}$; while agent $\mathcal{A}$ can output choice $\mathbf{X}$ since he / she thinks $\mathbf{P}$ is not similar enough to any choice. Based on that observation, we divide the 50 ChatGPT mis-aligned cases into two categories:

**Case I.** Human or ChatGPT fails to match the prediction with given choices and outputs an 'X'. **70%** misaligned samples belong to that case.

**Case II.** Human and ChatGPT successfully match the prediction with given choices, but the matching results are different. **30%** misaligned samples belong to that case.

In the two cases, **I** means the judgement of human and ChatGPT are less aligned (may due to different evaluation standards), while **II** means the judgement of human and ChatGPT is completely different. We manually investigate 15 samples in **Case-II**, and find that:

1. In 7 samples, ChatGPT did the right match while human did the wrong one;

2. In 6 samples, the model's prediction is ambiguous and related to multiple choices;

3. In 2 samples, human did the right match while ChatGPT did the wrong one. The results support that ChatGPT can have strong capability in choice matching, even when compared with human annotators. We visualize **Case-II** samples in Figure 6.

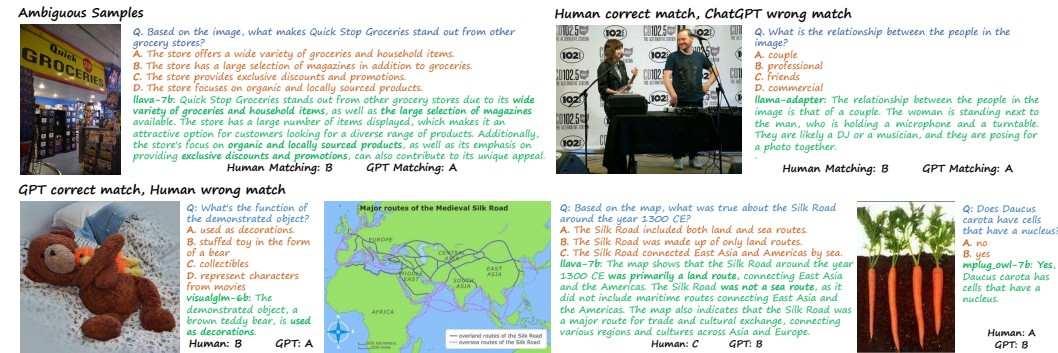

Figure 6: **Visualization of Case-II data samples.**

## D.2 CHATGPT-BASED EVALUATION FOR EXISTING MULTI-MODAL TASKS.

To demonstrate ChatGPT is a general evaluator, we also validate our ChatGPT-based evaluation paradigm on existing multi-modality tasks, including GQA (Hudson & Manning, 2019), OK-VQA (Marino et al., 2019), and Text-VQA (Singh et al., 2019). Given the ground-truth answer, we use GPT3.5 to score the VLM's prediction[3]. For each benchmark, we randomly select 1000 testing samples and evaluate with exact match (the traditional paradigm) and ChatGPT-based match, respectively, and list the results in Table 4. Basically, ChatGPT-based evaluation demonstrates the same trend compared to the exact-match accuracy on all tasks. On GQA, two algorithms demonstrate very close performance under ChatGPT-based evaluation. In further investigation, we find the reason is that ChatGPT succeeds in matching slightly different answers (compared to GT) generated by MiniGPT-4, while exact matching fails (examples in Table 5).

---

[3]The score will be an integer in [1, 2, 3, 4, 5]. 1 means completely wrong, while 5 means completely correct. We provide the prompt used for marking in Appendix.

Table 4: **ChatGPT-based marking** $vs$ **Exact Matching.** A preliminary study on VQA benchmarks. **Accuracy** is the success rate of answers being exactly matched with the groundtruth. For each sample, **GPT score** is an integer $n \in [1, 5]$, indicating the similarity between answer and groundtruth. We report the average GPT score for testing samples.

| Dataset | GQA (Hudson & Manning, 2019) | | OK-VQA (Marino et al., 2019) | | Text-VQA (Singh et al., 2019) | |
|---|---|---|---|---|---|---|
| Model | Flamingo | MiniGPT-4 | Flamingo | MiniGPT-4 | Flamingo | MiniGPT-4 |
| Accuracy | 33.6% | 22.4% | 42.6% | 21.9% | 22.9% | 9.8% |
| Average GPT score | 2.75 | 2.74 | 2.79 | 1.97 | 1.92 | 1.54 |

Many MiniGPT-4 GQA answers can not be matched with exact matching, but can be matched with ChatGPT-based evaluation. Here we provide some examples:

Table 5: MiniGPT-4 GQA answers (different from GT) that can be matched by ChatGPT-based evaluation.

| Question | GroundTruth | MiniGPT-4 Answer | GPT Score |
|---|---|---|---|
| On which side of the picture is the leather bag? | right | leather bag is on right side | 5 |
| Who is wearing a shirt? | girl | woman is wearing shirt | 5 |
| What is the pen made of? | plastic | pen is made of plastic## | 5 |
| What does the male person ride on? | bike | bicycle | 4 |
| Is the microwave silver and rectangular? | Yes | microwave is silver and rectangular | 5 |
| How does the silver lamp appear to be, on or off? | off | silver lamp appears to be off ## | 5 |

## E    DETAILS OF THE EVALUATED MODELS

In Table 6, we provide the details of the models evaluated in Section 4.2.

## F    ABILITY DIMENSIONS AND DATA SOURCE

In this section, we first introduce the definition of each leaf ability with more details. Then, we list all the data sources we adopted to build MMBench.

### COARSE PERCEPTION

1. **Image Style**: Determine which type of image it belongs to, such as photos, paintings, CT scans, etc.
2. **Image Scene**: Determine which environment is shown in the image, such as indoors, outdoors, forest, city, mountains, waterfront, sunny day, rainy day, etc.
3. **Image Emotion**: Determine which subjective emotion is conveyed by the overall image, such as cold, cheerful, sad, or oppressive.
4. **Image Quality**: Determine the objective quality of the image, such as whether it is blurry, bright or dark, contrast, etc.
5. **Image Topic**: Determine what the subject of the image is, such as scenery, portrait, close-up of an object, text, etc.

### FINE-GRAINED PERCEPTION (SINGLE-INSTANCE)

1. **Object Localization**: For a single object, determine its position in the image (such as top, bottom, etc.), its absolute coordinates in the image, count the number of objects, and the orientation of the object.
2. **Attribute Recognition**: Recognition of texture, shape, appearance characteristics, emotions, category.
3. **Celebrity Recognition**: Recognition of celebrities, landmarks, and well-known objects.

Table 6: **Details of the evaluated VLMs.**

| VLM | Language Backbone | Vision Backbone | Overall Parameters | Trainable Parameters |
|---|---|---|---|---|
| OpenFlamingo | LLaMA 7B | CLIP ViT-L/14 | 9B | 1.3B |
| OpenFlamingov2 | MPT 7B | CLIP ViT-L/14 | 9B | 1.3B |
| MMGPT | LLaMA 7B | CLIP ViT-L/14 | 9B | 22.5M |
| MiniGPT-4 | Vicuna 7B | EVA-G | 8B | 11.9M |
| MiniGPT-4-13B | Vicuna 13B | EVA-G | 14B | 11.9M |
| PandaGPT | Vicuna 13B | ImageBind ViT-H/14 | 14B | 28.8M |
| VisualGLM | ChatGLM 6B | EVA-CLIP | 8B | 0.2B |
| InstructBLIP | Vicuna 7B | EVA-G | 8B | 0.2B |
| InstructBLIP-13B | Vicuna 13B | EVA-G | 14B | 0.2B |
| Otter-I | LLaMA 7B | CLIP ViT-L/14 | 9B | 1.3B |
| LLaVA | LLaMA 7B | CLIP ViT-L/14 | 7.2B | 7B |
| LLaMA-Adapter | LLaMA 7B | CLIP ViT-L/14 | 7.2B | 1.2M |
| mPLUG-Owl | LLaMA 7B | CLIP ViT-L/14 | 7.2B | 0.4B |
| Shikra | LLaMA 7B | CLIP ViT-L/14 | 7.2B | 6.7B |
| IDEFICS-9B | LLaMA 7B | CLIP ViT-H/14 | 9B | 8.9B |
| IDEFICS-80B | LLaMA 65B | CLIP ViT-H/14 | 80B | 78.5B |
| Qwen-VL | Qwen 7B | ViT-G/16 | 9.6B | 9.6B |
| Qwen-VL-Chat | Qwen 7B | ViT-G/16 | 9.6B | 9.6B |

4. **OCR**: Recognition of text, formula, and sheet in the image.

FINE-GRAINED PERCEPTION (CROSS-INSTANCE)

1. **Spatial Relationship**: Determine the relative position between objects in image.

2. **Attribute Comparison**: Compare attributes of different objects in image, such as shape, color, etc.

3. **Action Recognition**: Recognizing human actions, including pose motion, human-object interaction, and human-human interaction.

ATTRIBUTE REASONING

1. **Physical Property Reasoning**: Predict the physical property of an object. Examples: he physical property of concentrated sulfuric acid is that it is volatile, the physical property of water is its fluidity, etc.

2. **Function Reasoning**: Predict the function of an object. Examples: the function of a broom is to sweep the floor, the function of a spatula is to cook, the function of a pen is to write, etc.

3. **Identity Reasoning**: Predict the identity of a person. Example: by observing a person's clothing and appearance, one may infer his / her occupation.

RELATION REASONING

1. **Social Relation**: Relations in human society or relations defined from the human perspective. Examples: Inter-person relations, such as father and son, husband and wife, friend, hostile, etc.

2. **Physical Relation**: All relationships that exist in the physical world, 3D spatial relationships and the connections between objects are.

3. **Nature Relation**: Other abstract relationships that exist in nature. Examples: predation, symbiosis, coexistence, etc.

LOGIC REASONING

1. **Structuralized Image-Text Understanding**: Structured understanding of images and text, including parsing the content of charts (such as the trends of multiple bars in a bar chart), understanding the code in an image, etc.

2. **Future Prediction**: Predict what will happen in the future. Examples: if it is thundering in the sky now, it can be predicted that it will rain soon (physical phenomenon); if someone raises their fist, it means they are going to hit someone (event occurrence); if someone's face becomes serious, it means they are going to get angry (emotional change).

Table 7: **The source of** $(Q, C, I, A)$ **in MMBench .** Customize/ChatGPT generated means this style of questions/choices are generated by the annotator/ChatGPT, given the image annotation.

| Image Source | Question Source | Choice and Answer Source |
|---|---|---|
| W3C School W3C (2023) | customize | matched code;unmatched code |
| Places Zhou et al. (2017) | customize | image-paired scene category;unpaired scene category |
| TextVQA Singh et al. (2019) | TextVQA | ground-truth answer;unpaired answer |
| ARAS Duan et al. (2022) | customize | image-paired action category;unpaired action category |
| CLEVR Johnson et al. (2017) | CLEVR | ground-truth answer;unpaired answer |
| PISC Li et al. (2017) | customize | image-paired social relation;unpaired social relation |
| KonIQ-10k Hosu et al. (2020) | customize | image-paired description;unpaired description |
| VSR Liu et al. (2023a) | customize | image-paired description;unpaired description |
| LLaVA Liu et al. (2023b) | ChatGPT generated | ChatGPT generated |
| COCO-Caption Chen et al. (2015) | customize | image-paired description;unpaired description |
| ScienceQA Lu et al. (2022) | ScienceQA | ground-truth answer;unpaired answer |
| Internet | customize | customize; customize |

# G ADDITIONAL EVALUATION RESULTS

In this section, we present additional MMBench evaluation results, including: 1) The performance of all VLMs on MMBench `dev` set, with accuracies of each L-2 capability reported; 2) the performance of VLMs on all 20 **L-3** abilities.

In Table 8, we present the evaluation results of all VLMs on MMBench `dev` set. Basically, the performance ranking is consistent with the MMBench `test` ranking, with only a few exceptions (the ranking is slightly different for IDEFICS-9B, mPLUG-Owl, *etc.*).

Table 8: **CircularEval results on MMBench `dev` set (L-2 abilities).** We adopt the following abbreviations: LR for Logical Reasoning; AR for Attribute Reasoning; RR for Relation Reasoning; FP-C for Fine-grained Perception (Cross Instance); FP-S for Fine-grained Perception (Single Instance); CP for Coarse Perception. The dash line separate models with the parameter size $\leq$ **10B** or $>$ **10B**.

| VLM | Overall | CP | FP-S | FP-C | AR | LR | RR |
|---|---|---|---|---|---|---|---|
| **OpenFlamingo (Alayrac et al., 2022)** | 4.6% | 2.0% | 6.8% | 2.8% | 8.0% | 6.8% | 0.0% |
| **OpenFlamingo v2 (Alayrac et al., 2022)** | 6.7% | 5.1% | 8.2% | 1.4% | 15.6% | 4.2% | 0.9% |
| **MMGPT (Gong et al., 2023)** | 15.2% | 20.6% | 13.7% | 3.5% | 26.6% | 2.5% | 13.0% |
| **MiniGPT-4 (Zhu et al., 2023)** | 32.0% | 44.3% | 40.6% | 13.3% | 35.7% | 14.4% | 13.9% |
| **InstructBLIP (Dai et al., 2023)** | 35.5% | 49.0% | 35.8% | 18.9% | 45.2% | 19.5% | 20.0% |
| **Qwen-VL (Bai et al., 2023)** | 38.2% | 46.6% | 35.2% | 39.2% | 44.7% | 16.1% | 34.8% |
| **VisualGLM (Du et al., 2022)** | 38.6% | 47.6% | 45.4% | 23.1% | 44.7% | 10.2% | 35.7% |
| **LLaMA-Adapter (Zhang et al., 2023b)** | 41.2% | 56.4% | 47.1% | 39.2% | 35.7% | 11.9% | 29.6% |
| **LLaVA (Liu et al., 2023b)** | 44.5% | 58.4% | 43.0% | 37.8% | 52.8% | 20.3% | 31.3% |
| **mPLUG-Owl (Ye et al., 2023)** | 48.1% | 58.8% | 51.9% | 44.8% | 50.8% | 16.1% | 43.5% |
| **IDEFICS-9B (Laurençon et al., 2023)** | 48.4% | 67.2% | 47.8% | 37.1% | 54.3% | 21.2% | 33.0% |
| **Otter-I (Li et al., 2023b;a)** | 51.6% | 65.9% | 46.4% | 39.2% | 57.3% | **32.2%** | 53.9% |
| **Shikra (Chen et al., 2023)** | 59.4% | 76.4% | 57.7% | **58.7%** | 57.3% | 26.3% | **58.3%** |
| **Qwen-VL-Chat (Bai et al., 2023)** | **60.6%** | **79.4%** | **66.2%** | 48.3% | 59.8% | **32.2%** | 43.5% |
| **PandaGPT (Su et al., 2023)** | 33.9% | 48.6% | 28.3% | 35.7% | 39.2% | 10.2% | 23.5% |
| **MiniGPT-4-13B (Zhu et al., 2023)** | 42.5% | 51.0% | 49.5% | 26.6% | 51.3% | 20.3% | 30.4% |
| **InstructBLIP-13B (Dai et al., 2023)** | 44.4% | 56.8% | 48.1% | 25.2% | 54.8% | 19.5% | 34.8% |
| **IDEFICS-80B (Laurençon et al., 2023)** | 54.8% | 64.5% | 58.4% | 44.8% | **65.8%** | 23.7% | 46.1% |

Table 9: **Results of leaf abilities (L-3) that belong to the Coarse Perception (CP) L-2 category.** Models are arranged in ascending order of **CP** top-1 accuracies.

| Split | VLM | CP | Image Style | Image Scene | Image Emotion | Image Quality | Image Topic |
|-------|-----|-----|-------------|-------------|---------------|---------------|-------------|
| DEV | **OpenFlamingo** | 2.0% | 0.0% | 1.9% | 0.0% | 1.9% | 8.3% |
| | **OpenFlamingo v2** | 5.1% | 3.8% | 3.8% | 4.0% | 3.8% | 13.9% |
| | **MMGPT** | 20.6% | 3.8% | 20.2% | 56.0% | 13.2% | 8.3% |
| | **MiniGPT-4** | 44.3% | 37.7% | 57.7% | 58.0% | 11.3% | 44.4% |
| | **Qwen-VL** | 46.6% | 30.2% | 78.8% | 28.0% | 5.7% | 63.9% |
| | **VisualGLM** | 47.6% | 37.7% | 74.0% | 50.0% | 0.0% | 52.8% |
| | **PandaGPT** | 48.6% | 30.2% | 70.2% | 62.0% | 1.9% | 63.9% |
| | **InstructBLIP** | 49.0% | 47.2% | 66.3% | 50.0% | 5.7% | 63.9% |
| | **MiniGPT-4-13B** | 51.0% | 64.2% | 62.5% | 56.0% | 3.8% | 61.1% |
| | **LLaMA-Adapter** | 56.4% | 45.3% | 75.0% | 76.0% | 3.8% | 69.4% |
| | **InstructBLIP-13B** | 56.8% | 73.6% | 74.0% | 58.0% | 11.3% | 47.2% |
| | **LLaVA** | 58.4% | 37.7% | 82.7% | 68.0% | 17.0% | 66.7% |
| | **mPLUG-Owl** | 58.8% | 67.9% | 61.5% | 82.0% | 18.9% | 63.9% |
| | **IDEFICS-80B** | 64.5% | 60.4% | 93.3% | 60.0% | 9.4% | 75.0% |
| | **Otter-I** | 65.9% | 64.2% | 85.6% | 70.0% | 17.0% | 77.8% |
| | **IDEFICS-9B** | 67.2% | 66.0% | 91.3% | 66.0% | 20.8% | 69.4% |
| | **Shikra** | 76.4% | 62.3% | **97.1%** | **86.0%** | 37.7% | **80.6%** |
| | **Qwen-VL-Chat** | **79.4%** | **83.0%** | 92.3% | 78.0% | **50.9%** | **80.6%** |
| TEST | **OpenFlamingo** | 1.5% | 0.0% | 4.7% | 0.0% | 1.2% | 0.0% |
| | **OpenFlamingo v2** | 4.1% | 1.2% | 5.4% | 3.6% | 2.4% | 7.1% |
| | **MMGPT** | 18.4% | 3.5% | 30.2% | 36.1% | 7.1% | 9.4% |
| | **Qwen-VL** | 36.4% | 23.5% | 60.5% | 13.3% | 0.0% | 71.8% |
| | **MiniGPT-4** | 36.6% | 32.9% | 63.6% | 33.7% | 2.4% | 36.5% |
| | **VisualGLM** | 41.3% | 30.6% | 68.2% | 41.0% | 0.0% | 52.9% |
| | **InstructBLIP** | 43.7% | 36.5% | 58.9% | 39.8% | 1.2% | 74.1% |
| | **PandaGPT** | 47.8% | 38.8% | 65.9% | 38.6% | 16.5% | 69.4% |
| | **MiniGPT-4-13B** | 50.7% | 62.4% | 75.2% | 38.6% | 4.7% | 60.0% |
| | **LLaMA-Adapter** | 50.7% | 31.8% | 70.5% | 44.6% | 24.7% | 71.8% |
| | **InstructBLIP-13B** | 54.2% | 61.2% | 72.1% | 42.2% | 5.9% | 80.0% |
| | **IDEFICS-9B** | 56.3% | 43.5% | 80.6% | 38.6% | 11.8% | 94.1% |
| | **LLaVA** | 59.3% | 55.3% | 82.9% | 51.8% | 15.3% | 78.8% |
| | **mPLUG-Owl** | 60.4% | 68.2% | 73.6% | 55.4% | 35.3% | 62.4% |
| | **Otter-I** | 60.8% | 62.4% | 82.9% | 47.0% | 4.7% | **95.3%** |
| | **IDEFICS-80B** | 62.1% | 55.3% | 85.3% | 48.2% | 16.5% | 92.9% |
| | **Shikra** | 71.9% | **72.9%** | **90.7%** | **60.2%** | 30.6% | **95.3%** |
| | **Qwen-VL-Chat** | **72.8%** | **72.9%** | 88.4% | 55.4% | **45.9%** | 92.9% |

In Tables 9 to 14, we present the performance of VLMs on all 20 **L-3** abilities. We noticed that the rankings on the **L-3** abilities are more diversified. Besides the leading VLMs Shikra and Qwen-VL-

Table 10: **Results of leaf abilities (L-3) that belong to the Fine-grained Perception (instance-level, FP-S) L-2 category.** Models are arranged in ascending order of **FP-S** top-1 accuracies.

| Split | VLM | FP-S | Object Localization | Attribute Recognition | Celebrity Recognition | OCR |
|---|---|---|---|---|---|---|
| DEV | OpenFlamingo | 6.8% | 2.5% | 10.8% | 4.0% | 15.4% |
| | OpenFlamingo v2 | 8.2% | 1.2% | 14.9% | 10.1% | 5.1% |
| | MMGPT | 13.7% | 1.2% | 8.1% | 27.3% | 15.4% |
| | PandaGPT | 28.3% | 14.8% | 52.7% | 23.2% | 23.1% |
| | Qwen-VL | 35.2% | 12.3% | 32.4% | 53.5% | 41.0% |
| | InstructBLIP | 35.8% | 6.2% | 50.0% | 45.5% | 46.2% |
| | MiniGPT-4 | 40.6% | 14.8% | 60.8% | 52.5% | 25.6% |
| | LLaVA | 43.0% | 12.3% | 64.9% | 50.5% | 46.2% |
| | VisualGLM | 45.4% | 22.2% | 43.2% | 67.7% | 41.0% |
| | Otter-I | 46.4% | 16.0% | 59.5% | 58.6% | 53.8% |
| | LLaMA-Adapter | 47.1% | 17.3% | 67.6% | 59.6% | 38.5% |
| | IDEFICS-9B | 47.8% | 14.8% | 56.8% | 68.7% | 46.2% |
| | InstructBLIP-13B | 48.1% | 14.8% | 56.8% | 69.7% | 46.2% |
| | MiniGPT-4-13B | 49.5% | 28.4% | 54.1% | 68.7% | 35.9% |
| | mPLUG-Owl | 51.9% | 19.8% | 60.8% | 76.8% | 38.5% |
| | Shikra | 57.7% | **32.1%** | **77.0%** | 63.6% | 59.0% |
| | IDEFICS-80B | 58.4% | 21.0% | 64.9% | 88.9% | 46.2% |
| | Qwen-VL-Chat | **66.2%** | 30.9% | 67.6% | **92.9%** | **69.2%** |
| TEST | OpenFlamingo | 2.5% | 1.9% | 3.0% | 2.5% | 2.6% |
| | OpenFlamingo v2 | 5.3% | 2.9% | 6.1% | 8.5% | 2.6% |
| | MMGPT | 17.8% | 2.9% | 22.2% | 28.0% | 16.9% |
| | PandaGPT | 20.1% | 7.7% | 36.4% | 19.5% | 16.9% |
| | Qwen-VL | 32.9% | 3.8% | 28.3% | 44.1% | 61.0% |
| | InstructBLIP | 34.4% | 2.9% | 42.4% | 40.7% | 57.1% |
| | MiniGPT-4 | 35.2% | 13.5% | 51.5% | 34.7% | 44.2% |
| | VisualGLM | 35.7% | 8.7% | 39.4% | 52.5% | 41.6% |
| | LLaVA | 37.9% | 10.6% | 45.5% | 39.8% | 62.3% |
| | IDEFICS-9B | 43.2% | 3.8% | 46.5% | 59.3% | 67.5% |
| | LLaMA-Adapter | 44.7% | 14.4% | 58.6% | 56.8% | 49.4% |
| | InstructBLIP-13B | 45.7% | 5.8% | 44.4% | 65.3% | 71.4% |
| | Otter-I | 46.7% | 11.5% | 53.5% | 57.6% | 68.8% |
| | MiniGPT-4-13B | 49.7% | 21.2% | 63.6% | 55.9% | 61.0% |
| | mPLUG-Owl | 53.0% | 16.3% | 62.6% | 78.0% | 51.9% |
| | IDEFICS-80B | 56.0% | 15.4% | 58.6% | 77.1% | 75.3% |
| | Shikra | 61.8% | **27.9%** | 71.7% | 70.3% | 81.8% |
| | Qwen-VL-Chat | **66.3%** | 20.2% | **74.7%** | **85.6%** | **88.3%** |

Chat, other VLMs, including IDEFICS-80B, LLaMA-Adapter, Otter-I, also achieve the first place in one / more sub-tasks on the `dev` / `test` split of MMBench.

Table 11: **Results of leaf abilities (L-3) that belong to the Fine-grained Perception (cross-instance, FP-C) L-2 category.** Models are arranged in ascending order of **FP-C** top-1 accuracies.

| Split | VLM | FP-C | Spatial Relationship | Attribute Comparison | Action Recognition |
|---|---|---|---|---|---|
| DEV | OpenFlamingo v2 | 1.4% | 2.2% | 2.3% | 0.0% |
| | OpenFlamingo | 2.8% | 2.2% | 0.0% | 5.6% |
| | MMGPT | 3.5% | 2.2% | 2.3% | 5.6% |
| | MiniGPT-4 | 13.3% | 4.4% | 15.9% | 18.5% |
| | InstructBLIP | 18.9% | 4.4% | 11.4% | 37.0% |
| | VisualGLM | 23.1% | 0.0% | 31.8% | 35.2% |
| | InstructBLIP-13B | 25.2% | 6.7% | 36.4% | 31.5% |
| | MiniGPT-4-13B | 26.6% | 20.0% | 20.5% | 37.0% |
| | PandaGPT | 35.7% | 11.1% | 25.0% | 64.8% |
| | IDEFICS-9B | 37.1% | 8.9% | 4.5% | 87.0% |
| | LLaVA | 37.8% | 2.2% | 36.4% | 68.5% |
| | Qwen-VL | 39.2% | 15.6% | 34.1% | 63.0% |
| | Otter-I | 39.2% | 15.6% | 4.5% | 87.0% |
| | LLaMA-Adapter | 39.2% | 11.1% | **47.7%** | 55.6% |
| | IDEFICS-80B | 44.8% | 24.4% | 13.6% | 87.0% |
| | mPLUG-Owl | 44.8% | 17.8% | 43.2% | 68.5% |
| | Qwen-VL-Chat | 48.3% | 20.0% | 31.8% | 85.2% |
| | Shikra | **58.7%** | **33.3%** | 45.5% | **90.7%** |
| TEST | OpenFlamingo v2 | 0.8% | 0.0% | 0.0% | 2.3% |
| | OpenFlamingo | 1.6% | 1.2% | 3.8% | 0.0% |
| | MMGPT | 5.3% | 3.7% | 3.8% | 8.0% |
| | MiniGPT-4 | 16.6% | 7.4% | 5.1% | 35.2% |
| | VisualGLM | 18.2% | 7.4% | 9.0% | 36.4% |
| | MiniGPT-4-13B | 19.8% | 17.3% | 9.0% | 31.8% |
| | PandaGPT | 20.6% | 12.3% | 15.4% | 33.0% |
| | InstructBLIP-13B | 24.3% | 9.9% | 17.9% | 43.2% |
| | InstructBLIP | 25.1% | 9.9% | 3.8% | 58.0% |
| | mPLUG-Owl | 25.1% | 25.9% | 17.9% | 30.7% |
| | Qwen-VL | 27.9% | 4.9% | 21.8% | 54.5% |
| | LLaVA | 30.8% | 11.1% | 26.9% | 52.3% |
| | LLaMA-Adapter | 33.6% | 19.8% | 28.2% | 51.1% |
| | Otter-I | 36.4% | 12.3% | 2.6% | 88.6% |
| | IDEFICS-9B | 36.8% | 6.2% | 9.0% | 89.8% |
| | Qwen-VL-Chat | 46.2% | 21.0% | **32.1%** | 81.8% |
| | IDEFICS-80B | 48.6% | **28.4%** | 21.8% | 90.9% |
| | Shikra | **50.6%** | 27.2% | 26.9% | **93.2%** |

Table 12: **Results of leaf abilities (L-3) that belong to the Attribute Reasoning (AR) L-2 category.** Models are arranged in ascending order of **AR** top-1 accuracies.

| Split | VLM | AR | Physical Property Reasoning | Function Reasoning | Identity Reasoning |
|---|---|---|---|---|---|
| DEV | OpenFlamingo | 8.0% | 10.7% | 8.9% | 2.2% |
| | OpenFlamingo v2 | 15.6% | 14.7% | 8.9% | 28.9% |
| | MMGPT | 26.6% | 24.0% | 10.1% | 60.0% |
| | MiniGPT-4 | 35.7% | 12.0% | 32.9% | 80.0% |
| | LLaMA-Adapter | 35.7% | 16.0% | 32.9% | 73.3% |
| | PandaGPT | 39.2% | 16.0% | 48.1% | 62.2% |
| | VisualGLM | 44.7% | 18.7% | 51.9% | 75.6% |
| | Qwen-VL | 44.7% | 32.0% | 53.2% | 51.1% |
| | InstructBLIP | 45.2% | 21.3% | 48.1% | 80.0% |
| | mPLUG-Owl | 50.8% | 18.7% | 59.5% | 88.9% |
| | MiniGPT-4-13B | 51.3% | 30.7% | 50.6% | 86.7% |
| | LLaVA | 52.8% | 33.3% | 53.2% | 84.4% |
| | IDEFICS-9B | 54.3% | 33.3% | 54.4% | 88.9% |
| | InstructBLIP-13B | 54.8% | 30.7% | 58.2% | 88.9% |
| | Otter-I | 57.3% | 29.3% | 63.3% | 93.3% |
| | Shikra | 57.3% | 30.7% | 64.6% | 88.9% |
| | Qwen-VL-Chat | 59.8% | 32.0% | **68.4%** | 91.1% |
| | IDEFICS-80B | **65.8%** | **46.7%** | 65.8% | **97.8%** |
| TEST | OpenFlamingo | 12.5% | 16.0% | 9.4% | 12.2% |
| | OpenFlamingo v2 | 12.8% | 9.0% | 7.5% | 24.4% |
| | MMGPT | 24.0% | 13.0% | 12.3% | 52.4% |
| | PandaGPT | 41.7% | 15.0% | 42.5% | 73.2% |
| | MiniGPT-4 | 42.4% | 29.0% | 36.8% | 65.9% |
| | Qwen-VL | 43.1% | 25.0% | 57.5% | 46.3% |
| | LLaMA-Adapter | 47.6% | 25.0% | 45.3% | 78.0% |
| | VisualGLM | 49.0% | 26.0% | 46.2% | 80.5% |
| | InstructBLIP | 49.7% | 28.0% | 49.1% | 76.8% |
| | mPLUG-Owl | 55.6% | 25.0% | 58.5% | 89.0% |
| | LLaVA | 57.6% | 36.0% | 54.7% | 87.8% |
| | InstructBLIP-13B | 59.7% | 24.0% | 67.9% | 92.7% |
| | MiniGPT-4-13B | 62.8% | 35.0% | 67.9% | 90.2% |
| | Otter-I | 63.5% | 45.0% | 61.3% | 89.0% |
| | IDEFICS-9B | 64.9% | 45.0% | 66.0% | 87.8% |
| | IDEFICS-80B | 68.1% | 35.0% | 79.2% | **93.9%** |
| | Shikra | 69.8% | 50.0% | 70.8% | 92.7% |
| | Qwen-VL-Chat | **74.3%** | **52.0%** | **81.1%** | 92.7% |

Table 13: **Results of leaf abilities (L-3) that belong to the Logic Reasoning (LR) L-2 category.** Models are arranged in ascending order of **LR** top-1 accuracies.

| Split | VLM | LR | Structuralized Image-Text Understanding | Future Prediction |
|---|---|---|---|---|
| DEV | **MMGPT** | 2.5% | 2.6% | 2.5% |
| | **OpenFlamingo v2** | 4.2% | 5.1% | 2.5% |
| | **OpenFlamingo** | 6.8% | 9.0% | 2.5% |
| | **VisualGLM** | 10.2% | 11.5% | 7.5% |
| | **PandaGPT** | 10.2% | 10.3% | 10.0% |
| | **LLaMA-Adapter** | 11.9% | 7.7% | 20.0% |
| | **MiniGPT-4** | 14.4% | 16.7% | 10.0% |
| | **Qwen-VL** | 16.1% | 19.2% | 10.0% |
| | **mPLUG-Owl** | 16.1% | 12.8% | 22.5% |
| | **InstructBLIP** | 19.5% | 17.9% | 22.5% |
| | **InstructBLIP-13B** | 19.5% | 19.2% | 20.0% |
| | **MiniGPT-4-13B** | 20.3% | 20.5% | 20.0% |
| | **LLaVA** | 20.3% | 19.2% | 22.5% |
| | **IDEFICS-9B** | 21.2% | 24.4% | 15.0% |
| | **IDEFICS-80B** | 23.7% | 24.4% | 22.5% |
| | **Shikra** | 26.3% | 16.7% | 45.0% |
| | Otter-I | **32.2%** | 20.5% | **55.0%** |
| | Qwen-VL-Chat | **32.2%** | **34.6%** | 27.5% |
| TEST | **MMGPT** | 1.2% | 0.0% | 2.8% |
| | **Qwen-VL** | 9.8% | 7.9% | 12.5% |
| | **OpenFlamingo** | 9.8% | 2.0% | 20.8% |
| | **OpenFlamingo v2** | 11.6% | 3.0% | 23.6% |
| | **VisualGLM** | 11.6% | 4.0% | 22.2% |
| | **MiniGPT-4** | 12.1% | 7.9% | 18.1% |
| | **LLaMA-Adapter** | 13.3% | 11.9% | 15.3% |
| | **LLaVA** | 15.0% | 8.9% | 23.6% |
| | **PandaGPT** | 15.6% | 6.9% | 27.8% |
| | **mPLUG-Owl** | 16.2% | 6.9% | 29.2% |
| | **MiniGPT-4-13B** | 17.3% | 6.9% | 31.9% |
| | **InstructBLIP-13B** | 17.3% | 5.9% | 33.3% |
| | **IDEFICS-9B** | 21.4% | 6.9% | 41.7% |
| | **InstructBLIP** | 22.0% | 4.0% | 47.2% |
| | **Otter-I** | 22.5% | 5.0% | 47.2% |
| | **IDEFICS-80B** | 29.5% | 12.9% | 52.8% |
| | **Shikra** | 33.5% | 13.9% | **61.1%** |
| | **Qwen-VL-Chat** | **40.5%** | **27.7%** | 58.3% |

Table 14: **Results of leaf abilities (L-3) that belong to the Relation Reasoning (RR) L-2 category.** Models are arranged in ascending order of **RR** top-1 accuracies.

| Split | VLM | RR | Social Relation | Physical Relation | Nature Relation |
|-------|-----|-----|-----------------|-------------------|-----------------|
| DEV | **OpenFlamingo** | 0.0% | 0.0% | 0.0% | 0.0% |
| | **OpenFlamingo v2** | 0.9% | 0.0% | 0.0% | 2.1% |
| | **MMGPT** | 13.0% | 14.0% | 0.0% | 18.8% |
| | **MiniGPT-4** | 13.9% | 27.9% | 0.0% | 8.3% |
| | **InstructBLIP** | 20.0% | 30.2% | 8.3% | 16.7% |
| | **PandaGPT** | 23.5% | 20.9% | 8.3% | 33.3% |
| | **LLaMA-Adapter** | 29.6% | 37.2% | 16.7% | 29.2% |
| | **MiniGPT-4-13B** | 30.4% | 53.5% | 8.3% | 20.8% |
| | **LLaVA** | 31.3% | 39.5% | 12.5% | 33.3% |
| | **IDEFICS-9B** | 33.0% | 46.5% | 16.7% | 29.2% |
| | **Qwen-VL** | 34.8% | 53.5% | 8.3% | 31.2% |
| | **InstructBLIP-13B** | 34.8% | 55.8% | 8.3% | 29.2% |
| | **VisualGLM** | 35.7% | 62.8% | 8.3% | 25.0% |
| | **Qwen-VL-Chat** | 43.5% | 58.1% | 25.0% | 39.6% |
| | **mPLUG-Owl** | 43.5% | 65.1% | 8.3% | 41.7% |
| | **IDEFICS-80B** | 46.1% | 62.8% | **29.2%** | 39.6% |
| | **Otter-I** | 53.9% | 76.7% | 20.8% | **50.0%** |
| | **Shikra** | **58.3%** | **93.0%** | 20.8% | 45.8% |
| TEST | **OpenFlamingo v2** | 1.4% | 1.3% | 3.8% | 0.0% |
| | **OpenFlamingo** | 3.3% | 0.0% | 11.5% | 1.2% |
| | **MiniGPT-4** | 14.2% | 11.5% | 19.2% | 13.6% |
| | **MMGPT** | 20.9% | 28.2% | 3.8% | 24.7% |
| | **InstructBLIP** | 22.3% | 30.8% | 11.5% | 21.0% |
| | **PandaGPT** | 22.3% | 28.2% | 9.6% | 24.7% |
| | **LLaMA-Adapter** | 23.2% | 32.1% | 19.2% | 17.3% |
| | **LLaVA** | 27.0% | 39.7% | 15.4% | 22.2% |
| | **VisualGLM** | 28.0% | 47.4% | 3.8% | 24.7% |
| | **IDEFICS-9B** | 29.4% | 51.3% | 11.5% | 19.8% |
| | **MiniGPT-4-13B** | 30.3% | 44.9% | 21.2% | 22.2% |
| | **Qwen-VL** | 30.3% | 46.2% | 9.6% | 28.4% |
| | **InstructBLIP-13B** | 36.5% | 48.7% | 25.0% | 32.1% |
| | **Otter-I** | 39.8% | 56.4% | 9.6% | 43.2% |
| | **mPLUG-Owl** | 40.3% | 62.8% | 21.2% | 30.9% |
| | **IDEFICS-80B** | 46.4% | 75.6% | 21.2% | 34.6% |
| | **Qwen-VL-Chat** | 47.9% | 60.3% | **30.8%** | **46.9%** |
| | **Shikra** | **53.1%** | **84.6%** | 19.2% | 44.4% |

# H MMBENCH DATA SAMPLES

In the Figures 7 to 12, we illustrate some examples in MMBench, grouped by the L-2 abilities.

**Image Style**

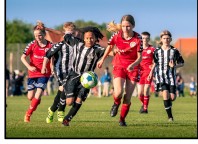

Q: Which category does this image belong to?
A. Oil Paiting
B. Sketch
C. Digital art
D. Photo
GT: A

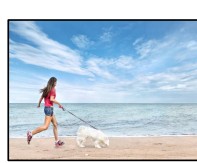

Q: Which category does this image belong to?
A. Oil Paiting
B. Sketch
C. Digital art
D. Photo
GT: B

**Image Topic**

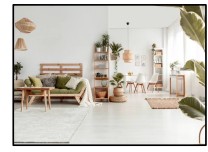

Q: Which of the following captions best describes this image?
A. A group of people playing soccer in a field
B. A woman walking her dog on a beach
C. A man riding a bicycle on a mountain trail
D. A child playing with a ball in a park
GT: A

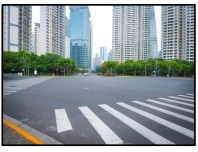

Q: Which of the following captions best describes this image?
A. A group of people playing soccer in a field
B. A woman walking her dog on a beach
C. A man riding a bicycle on a mountain trail
D. A child playing with a ball in a park
GT: B

**Image scene**

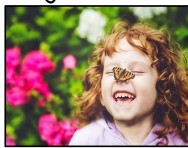

Q: What type of environment is depicted in the picture?
A. Home
B. shopping mall
C. Street
D. forest
GT: A

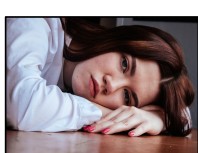

Q: What type of environment is depicted in the picture?
A. Home
B. shopping mall
C. Street
D. forest
GT: C

**Image Mood**

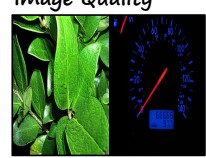

Q: Which mood does this image convey?
A. Cozy
B. Anxious
C. Happy
D. Angry
GT: C

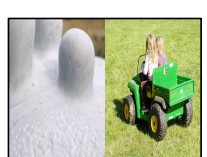

Q: Which mood does this image convey?
A. Sad
B. Anxious
C. Happy
D. Angry
GT: A

**Image Quality**

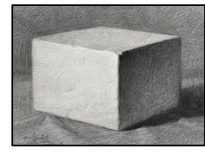

Q: Which image is more brightful?
A. The first image
B. The second image
GT: A

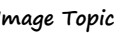

Q: which image is more colorful
A. The first image
B. The second image
GT: B

Figure 7: **Coarse Perception: Data samples.**

**Attribute Recognition**

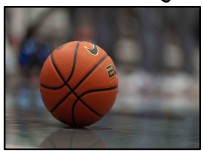

Q: What is the shape of this object?
A. Circle
B. Triangle
C. Square
D. Rectangle
GT: A

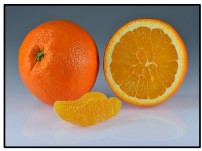

Q: what is the color of this object?
A. Purple
B. Pink
C. Gray
D. Orange
GT: D

**Celebrity Recognition**

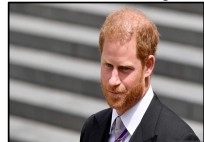

Q: Who is this person
A. David Beckham
B. Prince Harry
C. Daniel Craig
D. Tom Hardy
GT: B

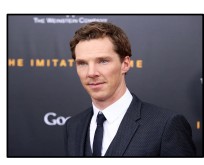

Q: Who is this person
A. Benedict Cumberbatch
B. Idris Elba
C. Ed Sheeran
D. Harry Styles
GT: A

**Object Localization**

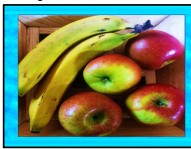

Q: How many apples are there in the image? And how many bananas are there?
A. 4 apples and 2 bananas
B. 3 apples and 3 banana
C. 2 apples and 4 bananas
D. 4 apples and 1 bananas
GT: A

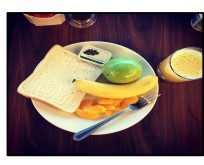

Q: Which corner is the juice?
A. Up
B. Down
C. Left
D. Right
GT: D

**OCR**

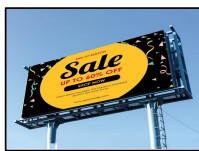

Q: What does this outdoor billboard mean?
A. Smoking is prohibited here.
B. Something is on sale.
C. No photography allowed
D. Take care of your speed.
GT: B

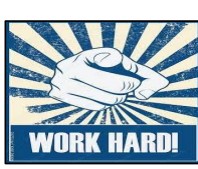

Q: What does this picture want to express?
A. We are expected to care for green plants.
B. We are expected to care for the earth.
C. We are expected to stay positive.
D. We are expected to work hard.
GT: D

Figure 8: **Fine-grained Perception (single-instance): Data samples.**

**Spatial Relationship**

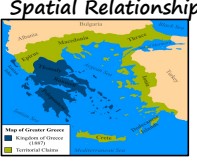

Q: Which sea is located in the south of Crete?
A. Ionian Sea
B. Aegean Sea
C. Black sea
D. Mediterranean Sea
GT: D

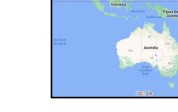

Q: What direction is Indonesia in Philippines?
A. northeast
B. southwest
C. southeast
D. northwest
GT: B

**Attribute Comparison**

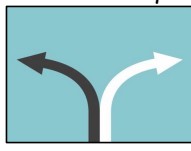

Q: Are the two arrows in the same direction in the picture?
A. Same
B. Not the same
C. Can't judge
GT: B

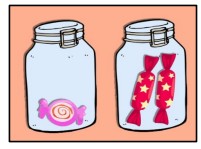

Q: Are the candies in the two jars in the picture the same color?
A. Same
B. Not the same
C. Can't judge
GT: B

**Action Recognition**

Q: What kind of human behavior does this picture describe?
A. A man with a solemn expression, XXX driving.
B. A man is practicing his skateboarding XXX skills.
C. A group of XXX breather from work.
D. A family is XXX clothing.
GT: A

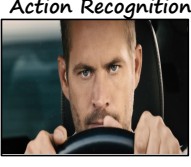

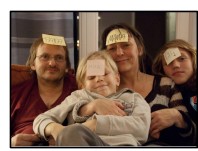

Q: What kind of human behavior does this picture describe?
A. This is a XXX smiles on their faces
B. A man is XXX his breathing and inner thoughts.
C. A musician XXX a classical piece.
D. A family is XXX together.
GT: A

Figure 9: **Fine-grained Perception (cross-instance): Data samples.** **XXX** indicates omitted contents which are less relevant to the question.

**Physical Property Reasoning**

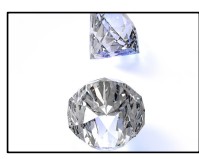

Q: The object shown in this figure:
A. Is the hardest naturally occurring substance on Earth.
B. Conducts electricity well at room temperature.
C. Is typically found in igneous rocks like basalt and granite.
D. Has a low melting point compared to other minerals.
GT: A

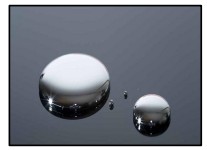

Q: The object shown in this figure:
A. Is one kind of metal that is liquid at the room temperature.
B. Can be easily dissolved in water.
C. Has a low boiling point compared to other metals.
D. Is attracted to magnets.
GT: A

**Function Reasoning**

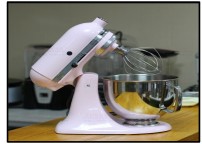

Q: What's the function of the demonstrated object?
A. Cut vegetables
B. stir
C. Water purification
D. Boiling water
GT: B

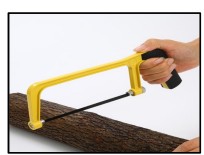

Q: What's the function of the demonstrated object?
A. Separating
B. Clamping
C. drill
D. incise
GT: A

**Identity Reasoning**

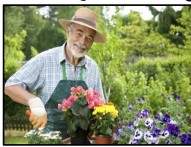

Q: What's the profession of the people in this picture?
A. Librarian
B. radio host
C. gardener
D. lawyer
GT: C

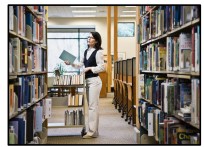

Q: What's the profession of the people in this picture?
A. Librarian
B. accountant
C. radio host
D. gardener
E. lawyer
GT: A

Figure 10: **Attribute Reasoning: Data samples.**

**Social_Relation**

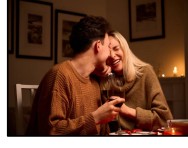

Q: What can be the relationship between the two persons in this image?
A. Father and daughter
B. Mother and son
C. Brother and sister
D. Husband and wife
GT: D

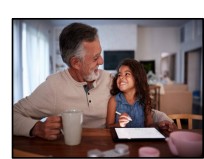

Q: What can be the relationship between the two persons in this image?
A. Father and daughter
B. Grandfather and granddaughter
C. Brother and sister
D. Husband and wife
GT: B

**Nature Relation**

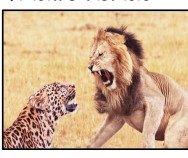

Q: In nature, what's the relationship between these two creatures?
A. Predatory relationships
B. Competitive relationships
C. Parasitic relationships
D. Symbiotic relationship
GT: B

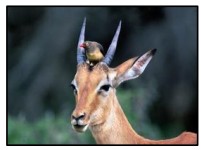

Q: In nature, what's the relationship between these two creatures?
A. Predatory relationships
B. Competitive relationships
C. Parasitic relationships
D. Symbiotic relationship
GT: D

**Physical Relation**

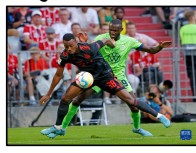

Q: Who is closer to the football in the image, the player in the black jersey or the player in the green jersey?
A. The player in the black jersey
B. The player in the green jersey
C. They are equally close
D. It cannot be determined
GT: A

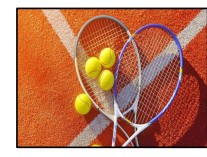

Q: How many tennis balls are placed on the tennis racket?
A. 1
B. 2
C. 3
D. 4
GT: C

Figure 11: **Relation Reasoning: Data samples.**

**Future Prediction**

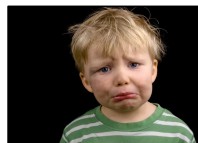

Q: What will happen next?
A.  this person is gonna cry
B.  this person is gonna laugh
C.  this person is gonna get mad
D.  both A,B, and C
GT: A

Q: What will happen next?
A.  the motorcyle is gonna go forward
B.  the motorcyle is gonna crash
C.  the motorcyle is gonna go
     backward
D.  both A,B, and C
GT: B

**Structuralized Image-text Understanding**

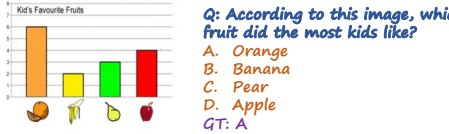

Q: According to this image, which
fruit did the most kids like?
A.  Orange
B.  Banana
C.  Pear
D.  Apple
GT: A

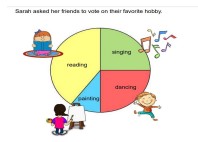

Q: According to this image, what
hobby is liked the least?
A.  Reading
B.  Singing
C.  Painting
D.  Dancing
GT: C

Figure 12: **Logic Reasoning: Data samples.**

