# OpenReview forum: "MMBench: Is Your Multi-modal Model an All-around Player?"
_ICLR.cc/2024/Conference — Submitted to ICLR 2024_

### Official Review · Reviewer_1JAs · 2023-10-27

**Soundness:** 3 good
**Presentation:** 2 fair
**Contribution:** 2 fair
**Rating:** 6
**Confidence:** 3

**Summary:**

This paper proposes a new benchmark MMBench for multimodal models. Unlike previous benchmarks suffering from scalable or bias problems, MMBench provides a comprehensive evaluation in an automatic ways. This benchmark reveals several drawbacks of current multimodal models, such as limited instruction-following and logic reasoning capabilities.

**Strengths:**

This work proposes a comprehensive benchmark and conducts extensive experiments on current multi-modal models. Additionally, this paper shows the bias of model's answer of singlechoice questions and proposes circular evaluation to improve the robustness.

**Weaknesses:**

The analysis and observations in this study align with prior research. Additionally, MME[1] highlights issues such as not following instructions, lack of reasoning and limited physical relation perception. Personally I encourage the presentation of novel insights regarding the shortcomings of existing MLLMs or suggestions for improvements.

While employing ChatGPT as the choice extractor can eliminate the need for manual ratings, it does introduce reliability concerns. Recent research[2] has revealed successful attacks to GPT by other LLMs, raising safety issues incorporating ChatGPT in evaluations.

[1] https://arxiv.org/pdf/2306.13394.pdf
[2] https://arxiv.org/pdf/2310.08419.pdf

**Questions:**

I am curious about the efficiency of this benchmark. As it uses CircularEval and incorporates chatGPT in the evaluation process, will it become much slower than other benchmarks?

---

> ### Author Response · Authors · 2023-11-17
>
> Thank you for your valuable advice. Here is our response to your questions:
>
> ## Q1: Insight brought by MMBench
>
> Numerous insights can be gleaned from the MMBench leaderboard, and we highlight some of the more apparent phenomena observed during the evaluation:
>
> 1. The performance of coarse perception is significantly lower than that of fine-grained perception across all models presented in Table 3 of the main paper. This trend is understandable, given that fine-grained perception necessitates the model's ability to precisely identify and comprehend the various objects present in an image. Furthermore, we discovered that enabling trainability of parameters in the vision encoder positively influences the model's fine-grained perception capabilities. This is because it equips the model with the flexibility to modify its vision feature extraction ability during visual-language pre-training and instruction tuning.
>
> 2. The performance of cross-instance finegrained perception is much lower than that of single-instance finegrained perception. The leaderboard of MMBench indicates that a majority of pre-training vision-language datasets primarily include descriptions or questions pertaining to individual objects, thereby neglecting to address cross-object relationships. Furthermore, we've observed that incorporating a referring expression comprehension task can effectively enhance the performance of the cross-instance fine-grained perception task.
>
> 3. The performance on perception tasks significantly surpasses that on reasoning tasks. Reasoning is a more complex task compared to perception, as it demands the model to not only accurately identify various objects in an image, but also possess the ability to reason about these objects, including their function and social identity. Therefore, enhancing the complexity of the instruction data, rather than posing simple questions about the color and shape of objects, is anticipated to improve this ability.
>
> ## Q2: ChatGPT is vulnerable to attack
>
> In the evaluation pipeline of MMBench, ChatGPT is solely utilized to extract answers from predictions and match them to the available choices, and the accuracy is calculated by some predefined functions based on the matching results, with CircularEval strategy. At present, we haven't identified any potential vulnerabilities that could affect ChatGPT within the evaluation pipeline and cannot think any potential attack approaches. If you have any ideas or suggestions on how to perform the attack, please feel free to propose them directly. We are more than willing to engage in a discussion with you.
>
> ## Q3: The efficiency of this benchmark.
>
> While exact matching has been utilized in numerous previous benchmarks like VQAv2 and GQA, incorporating ChatGPT into the evaluation pipeline may indeed slow down the process. However, when weighing the tradeoff between robustness and efficiency, using ChatGPT for extraction and matching emerges as the optimal choice. This is because it eliminates the issue of generating a high number of false negative samples, and the time required for evaluation remains within acceptable limits. The following table presents the number of ChatGPT API calles required by each VLM under VanillaEval and CircularEval.
>
> | Model            | VanillaEval GPT Calls | CircularEval GPT Calls |
> | ---------------- | --------------------- | ---------------------- |
> | OpenFlamingov2   | 13                    | 7                      |
> | LLaVA-7B         | 993                   | 1908                   |
> | VisualGLM-6B     | 1051                  | 1990                   |
> | MMGPT-9B         | 143                   | 229                    |
> | InstructBLIP-13B | 69                    | 102                    |
> | mPLUG-Owl-7B     | 701                   | 1510                   |
> | Qwen-VL          | 245                   | 352                    |
> | MiniGPT-4-7B     | 656                   | 1026                   |
> | Idefics-80B      | 23                    | 36                     |
> | PandaGPT         | 1167                  | 1993                   |
> | MiniGPT-4-13B    | 524                   | 1120                   |
> | OpenFlamingo     | 68                    | 57                     |
> | Llama-adapter    | 960                   | 1624                   |
> | Otter-9B         | 0                     | 0                      |
>
> Among all VLMs, PandaGPT consumes most ChatGPT API calls (\~ 2000). We find that each answer matching prompt is 500 tokens length on average. Based on the current price of ChatGPT (GPT-3.5, which is 0.001 USD / 1k tokens), the price upper bound for each evaluation is only 1 USD  (\~0.3 USD for all models on average). In practice, we find that for most models, the evaluation can finish in half an hour, with one single OpenAI key and parallel API calling.

---

> > ### Author Response · Authors · 2023-11-20
> > **Looking forward to your reply!**
> >
> > We sincerely appreciate your great efforts in reviewing this paper. Your constructive advice and valuable comments really help improve our paper. Considering the approaching deadline, please, let us know if you have follow-up concerns. We sincerely hope you can consider our reply in your assessment, and we can further address unclear explanations and remaining concerns if any.
> >
> > Once more, we are appreciated for the time and effort you've dedicated to our paper.

---

> > ### Comment · Reviewer_1JAs · 2023-11-20
> >
> > Thanks for your reply. It address my questions and I raise my rating to 6.

---

> > > ### Author Response · Authors · 2023-11-21
> > >
> > > Thank you for adjusting the review score!! We're pleased that your concerns have been addressed.

---

### Official Review · Reviewer_g871 · 2023-10-27

**Soundness:** 1 poor
**Presentation:** 2 fair
**Contribution:** 2 fair
**Rating:** 3
**Confidence:** 4

**Summary:**

The paper presents a new multiple-choice VQA evaluation benchmark for assessing recent multimodal language large models without subjective human evaluation. The benchmark is set up to evaluate the perception and reasoning abilities of these models, such as attribute prediction, OCR, action recognition, social relation, and so on. It currently consists of 2948 single choice questions covering over 20 abilities. It comprehensively reports the performance of recent 18 models including LLaVA, InstructBLIP, Shikra, and etc.

**Strengths:**

There are several strengths about this work:
- The vision-language community certainly needs more objective benchmarks for evaluating recent multimodal models.
- The proposed VQA benchmark covers a wide array of abilities (over 20).
- The paper comprehensively tests most recent multimodal models (18 of them).

**Weaknesses:**

I have several major concerns about dataset collection and evaluation strategies.

> **Dataset Collection and Quality**

As the major contribution of this paper is the new VQA benchmark, I find the paper did a **poor job in explaining how the samples are generated, collected, and verified**. For example, how did you select images from existing sources? How did the annotator come up with QA pairs based on the images? How did you verify the correctness/relevance of these samples? From the current paper it is really hard to gauge the data quality of the benchmark.

After I downloaded the public dev set, I can easily find a lot of VQA samples across categories that can be **solved without looking at the images**. Here are some examples:

>**Example 1**

Image: a photo of an African elephant.

Question: The African elephant is the () land animal in the world.

Options: (A) smallest, (B) largest.

Category: attribute_recognition.

> **Example 2**

Image: a photo of a snow rabbit.

Question: Which animal’s skin is also adapted for survival in cold places?

Options: (A) fantastic leaf-tailed gecko, (B) polar bear.

Category: physical_property_reasoning.

> **Example 3**

Image: a photo of the world map centered on Australia.

Question: What direction is Indonesia in Australia?

Options: (A) northwest, (B) northeast, (C) southwest, (D) southeast.

Category: spatial_relationship

Even though I cannot attach images to my review, it is clear that these questions can be answered by a human without looking at the images. This makes MMBench more like a QA instead of a VQA benchmark. The authors should discuss how MMBench is collected and why such problematic samples can leak into your dev set. Did you use crowd-sourced or expert annotators? This is an important question to answer especially if MMBench continues to expand -- how do you plan to ensure the quality of collected samples?

Finally, the paper did not discuss the **licensing for collected Internet images**. I also find some images of MMBench containing **watermarks** from existing website such as zhihu.com.

> **Evaluation Strategy**

I am also concerned with the CircularEval strategy. In section 3.1, the paper says *“CircularEval doesn’t necessarily require N x inference cost”* because if the VLM makes a wrong prediction in one pass, the following passes can be dropped. As such, the paper claims this strategy has *“an affordable cost“* (Section 3). I find this to be a very misleading statement because the computation cost is indeed O(N) and a “perfect” model will still require N passes.

Could you explain why not doing N passes then report the average accuracy?

Human performance on MMBench is currently missing. This is important to gauge the overall difficulty of this benchmark.

Finally, answer extraction using LLMs is a standard practice in NLP [1], and thus it is hardly a novel contribution.

> **References**

[1] Large Language Models are Zero-Shot Reasoners. Kojima et al. 2022.

**Questions:**

In addition to my questions listed above in weakness section, I have one extra question regarding **related work**:

Could you explain why MME has *"a non-rigorous evaluation setting"* that makes it harder to reveal the real performance gap between VLMs? (Appendix A.1)

**Details Of Ethics Concerns:**

The paper did not discuss the licenses/copyright of their Internet collected images. I can see some images in their dataset including watermark from websites such as zhihu.com.

---

> ### Author Response · Authors · 2023-11-17
>
> Thank you for your valuable advice. Here is our response to your questions:
>
> ## Q1: Quality of the benchmark
>
> ### Q1.1: How are these samples collected?
>
> First, we recruit a group of college students as volunteers. We then provide necessary training to these volunteers, equipping them with skills to collect images and create relevant questions and answers for each task. Moreover, we provide them with a range of potential sources to gather images and formulate corresponding questions and choices. Table 7 in the paper showcases these potential sources. Following this, the collected Visual Question Answering (VQA) data undergoes a quality inspection pipeline to ensure the creation of a reliable benchmark.
>
> ### Q1.2: Quality testing for samples in MMBench
>
> We conduct thorough quality testing on the collected samples through manual review, with a primary focus on two key aspects:
> 1. Ensuring the accuracy and coherence of the questions, without any logical or factual errors.
> 2. Verifying the presence of a single correct answer that is pertinent to the question.
> We filter out any samples that fail to adhere to these aforementioned criteria.
>
> ### Q1.3: Question can be answered without the corresponding image
>
> Thank you for highlighting this issue. We too identified the same problem after submitting our paper. To determine the number of questions in MMBench that can be answered without an image, we input all the questions into GPT-4 and ensure that it makes reasonable guesses for all questions. We discovered that ~13% of the dev questions and ~10% of the test questions could be answered correctly, which mainly originated from ScienceQA. Subsequently, we removed these questions and used the remaining ones to evaluate all models. As indicated in the table below, the performance of these models remained similar to their performance prior to the image filter, and the rankings of the different models were basically consistent. That suggests that the current MMBench leaderboard remains credible. Furthermore, we plan to release an updated version of MMBench, with the questions that can be answered without images filtered out.
>
> | Model            | Dev   | Dev w/o. GPT-4 | Test  | Test w/o. GPT-4 |
> | ---------------- | ----- | -------------- | ----- | --------------- |
> | Qwen-VL-Chat     | 60.57 | 59.9           | 61.83 | 60.75           |
> | Shikra           | 59.36 | 60             | 60.43 | 60.81           |
> | IDEFICS-80B      | 54.81 | 51.29          | 54.82 | 52.34           |
> | IDEFICS-9B       | 48.37 | 47.03          | 45.52 | 44.8            |
> | mPLUG-Owl        | 48.11 | 48.51          | 46.41 | 46.11           |
> | LLaVA            | 44.5  | 42.77          | 42.21 | 41.74           |
> | InstructBLIP-13B | 44.42 | 43.17          | 43.39 | 43.18           |
> | MiniGPT-4-13B    | 42.53 | 40.5           | 42.54 | 41.62           |
> | LLaMA-Adapter    | 41.24 | 41.09          | 39.63 | 39.63           |
> | VisualGLM        | 38.57 | 37.82          | 33.63 | 33.52           |
> | Qwen-VL          | 38.23 | 36.53          | 32.23 | 31.34           |
> | InstructBLIP     | 35.48 | 33.66          | 35.37 | 34.08           |
> | PandaGPT         | 33.93 | 32.77          | 30.72 | 29.1            |
> | MiniGPT-4        | 32.04 | 30.69          | 29.43 | 28.04           |
> | MMGPT            | 15.21 | 13.66          | 15.98 | 15.26           |
> | OpenFlamingo v2  | 6.7   | 5.45           | 5.72  | 4.61            |
> | OpenFlamingo     | 4.64  | 3.37           | 4.54  | 3.55            |
>
> ### Q1.4  Licensing for collected Internet images
>
> All images within MMBench are available for academic use.  During the collection process, we instruct our annotators to document the original URL of each image. Subsequently, we manually review these URLs individually, eliminating any images that are not suitable for academic use. Moreover, should any authors request the removal of their images from MMBench, we will promptly comply.
>
> ### Q1.5 MMBench is a QA benchmark instead of a VQA one
>
> The conclusion seems overly severe to be considered fair. As demonstrated in Q1.3, a mere 10% of the existing questions can be answered without the aid of images. The remaining majority of questions necessitate the model's accurate comprehension of the image content to provide correct responses.

---

> > ### Author Response · Authors · 2023-11-17
> >
> > ## Q2: CircularEval
> >
> > 1. The inference cost in this context is solely concerned with the cost associated with querying ChatGPT, and does not take into account the cost of generating predicted answers from a vision-language model.
> > 2. A single question will be input into a vision-language model N times, with the choices being shifted each time (refer to Figure 3 in the main paper for more details). CircularEval will only assign a success score to the current question if the model correctly answers the question all N times. If the model provides an incorrect answer even once out of the N times, CircularEval will immediately assign a fail score. This eliminates the need to query ChatGPT for the remaining samples, thereby significantly reducing the inference cost. For instance, if we utilize VanillaEval, we will, on average, prompt ChatGPT 383 times for each model. However, with CircularEval, this number rises to 681. This represents a 78% increase in querying ChatGPT, which is significantly less than a 300% increase. The following table shows the times to query ChatGPT for each model using VanillaEval and CircularEval.
> >
> > | Model | VanillaEval GPT Calls | CircularEval GPT Calls |
> > | -- | -- | -- |
> > | OpenFlamingov2   | 13   | 7 |
> > | LLaVA-7B         | 993 | 1908 |
> > | VisualGLM-6B     | 1051 | 1990 |
> > | MMGPT-9B         | 143 | 229 |
> > | InstructBLIP-13B | 69 | 102 |
> > | mPLUG-Owl-7B     | 701 | 1510 |
> > | Qwen-VL          | 245 | 352 |
> > | MiniGPT-4-7B     | 656 | 1026 |
> > | Idefics-80B      | 23 | 36|
> > | PandaGPT         | 1167| 1993|
> > | MiniGPT-4-13B    | 524| 1120|
> > | OpenFlamingo     | 68| 57|
> > | Llama-adapter    | 960 | 1624 |
> > | Otter-9B         | 0 | 0 |
> >
> > 3. Another interesting fact is, sometimes CircularEval even requires less ChatGPT calls compared to VanillaEval. This happens when the heuristic strategy failed to match the VanillaEval prediction with any options, but succeeded in matching one of the CircularEval predictions with a wrong option.
> > 4. For an perfect model capable of producing the label of a choice (A, B, C, D), we can directly match the prediction with the target answer, eliminating the need to query ChatGPT. In experiments, we do not find a single VLM with both perfect multi-modal understanding performance and extremely poor instruction-following capability (to output ABCD).
> >
> > ## Q3: Answer extraction using LLMs is standard practice in NLP
> >
> > 1. In the evaluation process, we first use heuristic matching to match the option label (A, B, C, D) or the option content with the model's predictions. Only if the matching failed, we utilize ChatGPT to align the semantic meaning of the prediction with one of the available choices, subsequently outputting the label for the corresponding choice. Such work is more complicated compared to straight-forward answer matching.
> > 2. To the best of our knowledge, we are the first to integrate the strong semantic matching capability of cutting-edge LLMs into the evaluation process of multimodal understanding. It makes the evaluation procedure yield more accurate results and reduces the number of false negatives.
> >
> > ## Q4: MME uses a non-rigorious evaluation setting
> > 1. MME continues to utilize the traditional method of exact matching, which necessitates the model to produce precise 'yes' or 'no' responses. This approach, however, tends to generate a number of false negative samples.
> > 2. The metric adopted by MME (accuracy & accuracy +) enables randomd guessing to achieve a substantial amout of score (37.5%), and also makes it difficult to reveal the performance gap between VLMs. Besides, It appears unreasonable that certain models yield scores on the MME that are lower than those of random guesses (especially for perception tasks). The table below provides a comprehensive overview of this issue. Notably, MMBench does not suffer from this problem. Some examples:
> >     1. XComposer outperforms Qwen-VL-Chat by 14.2% Top-1 accuracy on MMBench-Dev, which is a substantial gap. However, in MME, the gap is merely 70 in total score (2.5% of the total score).
> >     2. In MME, VisualGLM-6B and MiniGPT-4 demonstrates poorer performance compared to the random baseline, which doesn't sounds reasonable. However, in MMBench-Dev, the two approaches significantly outperforms the random baseline (which is <0.5% Top-1 accuracy).
> >
> > | Method       | MME-Perception | MME-Cognition | MME Total | MMBench-Dev |
> > | ------------ | -------------- | ------------- | --------- | ----------- |
> > | GPT4-V       | 1409           | 517           | 1926      | 75.1%       |
> > | XComposer    | 1528           | 391           | 1919      | 74.8%       |
> > | Qwen-VL-Chat | 1488           | 361           | 1849      | 60.6%       |
> > | mPLUG-Owl2   | 1450           | 313           | 1763      | 66.5%       |
> > | VisualGLM-6B | 705            | 182           | 887       | 38.1%       |
> > | MiniGPT-4    | 582            | 144           | 726       | 24.3%       |
> > | Random       | 750            | 375           | 1225      | <0.5%       |

---

> > > ### Author Response · Authors · 2023-11-20
> > > **Looking forward to your reply!**
> > >
> > > We sincerely appreciate your great efforts in reviewing this paper. Your constructive advice and valuable comments really help improve our paper. Considering the approaching deadline, please, let us know if you have follow-up concerns. We sincerely hope you can consider our reply in your assessment, and we can further address unclear explanations and remaining concerns if any.
> > >
> > > Once more, we are appreciated for the time and effort you've dedicated to our paper.

---

> ### Comment · Reviewer_g871 · 2023-11-20
> **Reply to authors**
>
> I found the author responses to my major concern (dataset quality / curation) to be vague.
>
> > **Response: ...college students as volunteers. We then provide necessary training to these volunteers, equipping them with skills to collect images and create relevant questions and answers for each task...**
>
> What does it mean by **necessary training** and **skills to collect relevant questions**? I do not find details regarding this important information in the updated version of the paper. Please update the paper with the detailed procedure.
>
> > **Response: ..we input all the questions into GPT-4 and ensure that it makes reasonable guesses for all questions. We discovered that ~13% of the dev questions and ~10% of the test questions could be answered correctly, which mainly originated from ScienceQA. ... Furthermore, we plan to release an updated version of MMBench, with the questions that can be answered without images filtered out.**
>
> How did you use ChatGPT for reasonable guesses? And do you use CircularEval? What is the exact prompt you used for conversing with ChatGPT? I do not find this discussion in the updated paper. Also, given that MMBench is already used in many recent MLLM evaluation, how do you plan to update this benchmark? Do you plan to release a MMBench-2, or simply request all previous works who used the earlier version of MMBench to re-evaluate their models?
>
> > **The author did not discuss the watermark issue in their response.**
>
> > **I found the author's response regarding to CircularEval to be confusing.**
>
> Reading the response, I believe the term used in paper (**CircularEval doesn’t necessarily requires N× inference cost.**) is just misleading. **Inference cost** usually refers to the cost of model inference. The authors should update the paper to clarify that this has to do with "ChatGPT calls/querying cost".
>
> > **Response: sometimes CircularEval even requires less ChatGPT calls compared to VanillaEval. This happens when the heuristic strategy failed to match the VanillaEval prediction with any options, but succeeded in matching one of the CircularEval predictions with a wrong option.**
>
> Please clarify this with a concrete example. I find this to be confusing.
>
> > **Thank you for clarifying the issues of MME. I think this discussion should go into the updated paper to justify why people should use your benchmark instead of MME."**

---

> > ### Author Response · Authors · 2023-11-21
> >
> > We appreciate your suggestion and have subsequently updated crucial information in the paper (highlighted in red). To alleviate any confusion or misunderstandings, we will try our best to address each question in the sections below.
> >
> > ## Q1: Necessary training & skills to collect relevant questions.
> > The relevant details are updated in Sec 2.2 of the updated paper.
> > ## Q2: exact prompt for ChatGPT and the plan to update benchmark.
> > 1. **Evaluation LLM-based Approaches** We use the following prompt for LLMs (GPT-3.5, GPT-4, Claude-2), which instructs them to try their best to solve the provided question, even without the associated image. CircularEval also applies to the LLM-based pure-text evaluation. More relevant details are updated in Appendix.B of the paper.
> > ```python
> > prompt_tmpl = """
> > You are an AI assistant which is designed to answer questions from people.
> > You will be asked a question and provided several choices, and you need to choose the best choice to answer the question.
> > There is an image associated with the problem, but that is not provided to you, so you need to try your best to hallucinate the image.
> > Again, you can only choose from the provided choices, and output a single uppercase letter (A, B, C, D, etc.). \n
> > Question: <question begins> {} <question ends>; \n
> > Choices: <choices begin> {} <choices end>. \n
> > Your answer:
> > """
> > example_question = "How many apples are there in the image?"
> > option_str = "A. 4\nB. 3\nC. 2\nD. 1"
> > prompt = prompt_tmpl.format(example_question, option_str)
> > ```
> > 2. **Update Plan** Note that the questions can not be resolved purely based on the language part are a subset of the original dataset. Since we have saved the inference results of all VLMs we tested, we don't need the model authors to resubmit the evaluation result. To update the benchmark, we plan to provide a new dataset version (MMBench v1.1, if we say the current version is MMBench v1.0) and calculate the accuracies on both dataset versions. When a new user creates a new submission, he / she can see the results on both dataset versions. For all previous submissions which are close source, we will ask for the authors' permission first before disclosing the performance on the new dataset version.
> > ## Q3: watermark issue
> > The issue of watermarking is analogous to that of licensing, which we have discussed in Q1.4 above. Specifically, regarding the watermark associated with Zhihu, we have consulted their copyright policy. It prohibits the misuse of their content for commercial purposes, but does not restrict its use for academic and non-profit purposes.
> > ## Q4: CircularEval "Inference Cost"
> > Thanks for your suggestion, we have updated the related term in Sec 3.1.

---

> > ### Author Response · Authors · 2023-11-21
> >
> > ## Q5: Sometimes CircularEval even requires less ChatGPT calls compared to VanillaEval.
> > Below is an example in which CircularEval even requires less ChatGPT calls compared to VanillaEval:
> > Assume the following inputs:
> > 1. (Vanilla / Circular-1) **Q**: What is the main object in the image? **Options**: A. a teddy bear; B. a rabbit; C. a cat; D. a dog. **Prediction**: a cute little rabbit. **GroundTruth**: A.
> > 2. (Circular-2) **Q**: What is the main object in the image? **Options**: A. a rabbit; B. a cat; C. a dog; D. a teddy bear. **Prediction**: D. **GroundTruth**: D.
> > 3. (Circular-3) **Q**: What is the main object in the image? **Options**: A. a cat; B. a dog; C. a teddy bear; D. a rabbit. **Prediction**: D. **GroundTruth**: C.
> > 4. (Circular-4) **Q**: What is the main object in the image? **Options**: A. a dog; B. a teddy bear; C. a rabbit; D. a cat. **Prediction**: B. a teddy bear. **GroundTruth**: B.
> >
> > (1). When performing VanillaEval:
> > a. Try to match "a cute little rabbit" with the content / option label of each option, failed.
> > b. Call ChatGPT to extract the option label from the prediction given the options.
> > **ChatGPT call count: 1**
> >
> > (2). When performing CircularEval:
> > **a**. For each circular pass, we try to extract the options labels from the predictions:
> >     i. Circular-1: Failed to extract.
> >     ii. Circular-2: Extracted: D; Groundtruth: D.
> >     iii. Circular-3: Extracted: D; Groundtruth: C.
> >     iv. Circular-4: Extracted: B; Groundtruth: B.
> > **b**. In Circular-3, we find that the extracted option label is not the same as the groundtruth. Under CircularEval, the VLM has already failed this question. So not ChatGPT call is required to extract the option label from Circular-1 prediction.
> > **ChatGPT call count: 0**
> >
> > That's the case when CircularEval requires less ChatGPT calls compared to VanillaEval.
> >
> > ## Q6： Comparison with MME
> > Thank you for your suggestion, and we have made revisions to the  paper on the discussion about MME in Appendix C.
> > Meanwhile, we would like to clarify that we do not want to justify why people should use MMBench instead of MME. MME is a significant contemporary work alongside MMBench, both concentrate on evaluating different facets of a vision-language model. Being one of the first benchmarks to evaluate the versatile capabilities of VLMs, MME has great impact on the vision-language modeling community. Meanwhile, MMBench proposes a benchmark with advanced answer extraction strategy and more rigorous evaluation metrics, thus it also has its own merit. We believe that each benchmark can complement the other, working together to advance the development of vision-language models.

---

> ### Comment · Reviewer_g871 · 2023-11-22
> **Thank you for the reply**
>
> I carefully checked the updated paper. I appreciate that the authors took most of reviewers' feedbacks into consideration for improving the draft. I have a remaining question regarding the use of public data sources.
>
> > **Quote from paper: "Firstly, our data is gathered from the validation sets of these public datasets, not their training sets. Secondly, data samples procured from these public datasets constitute less than 10% of all MMBench data samples."**
>
> I think these statements can be misleading -- I saw that LLaVA is one of the data sources for constructing MMBench; however it does not come with an official validation set (unless you are referring to the 30-images LLaVA-Bench). Also, I do not find the actual distribution of data sources in the current draft.

---

> > ### Comment · Reviewer_g871 · 2023-11-22
> > **Some last comments**
> >
> > I still think the current evaluation protocol, especially when coupled with both heuristic matching and ChatGPT answer extraction strategy, is not conceptually simple and may lead to unfair comparison. For example, if outputting A/B/C/D is challenging because current VLMs lacks instruction-following abilities, a new VLM/prompt engineer can intentionally choose to ignore the multiple choices and directly output the final answer, thus leaving all answer extraction to the powerful ChatGPT. Also, answer extraction does not seem trivial for this dataset as even the SOTA open-source LLaMA fail to align with human (compared against ChatGPT).
> >
> > I think for such small-scale (no more than 10K samples) manually-constructed benchmark, the dataset quality is the most important factor because even including a few biased samples could potentially affect the future development of VLMs. While it is not possible for reviewers to perform manual check of the dataset quality, I think the authors should release comprehensive training materials and open-source the platform such that future work can follow the same protocol to collect samples. I will consider this to be a more important scientific contribution than the dataset itself.

---

> ### Author Response · Authors · 2023-11-23
> **Thanks for the update**
>
> # Our Responses:
>
> 1. **LLaVA**:
>
>     Apologies for any confusion caused. When we talk about "the validation sets", it only applies to datasets that have both
>     "training" and "validation" splits (like COCO or ScienceQA). For LLaVA, we utilize the images in the full set. We will update the
>     paper accordingly to resolve the confusion.
>
> 2. **Data Source Statistics**:
>
>     Since the discussion deadline is approaching, we may not be able to provide the source statistics at this time (some raw data
>     need to be processed to figure out the exact source of each image). We promise that the detailed statistics will be included in the
>     next version of the manuscript.
>
> 3. **Answer Extraction**:
>
>     3.1. __ChatGPT-based Answer Extraction__:
>
>     Most VLMs and LLMs are designed to **chat** with human beings: If not finetuned with a sufficient number of multiple-choice
>     problems, most of them can not output the option label for every question perfectly (ie., follow the instruction of solving multiple-
>     choice problems). Our ChatGPT-involved evaluation pipeline is designed to evaluate VLMs solely based on the answer content,
>     and we think that pipeline can be more fair compared to exact matching.
>
>     If we use exact matching during the evaluation, VLMs finetuned on a large number of multiple-choice problems will follow the
>     instructions in MMBench much better, and significantly outperforms VLMs not finetuned on multiple-choice problems. Then it will
>     be difficult to reveal the **real** multi-modal understanding performance gap with the evaluation results.
>
>     3.2. __Poor performance of LLaMA__:
>
>     Actually, the overall performance of Vicuna (v1.0, finetuned based on LLaMA) is not **SOTA**, especially at the current time
>     (2023.11), thus it performs poorly on MMBench answer extraction. In the upcoming version, we will try to utilize more advanced
>     opensource LLMs (XWinLM, UltraLM, Vicuna v1.5, etc.) to see if they can achieve better performance on MMBench answer
>     extraction. If so, we will have a ChatGPT free evaluation pipeline.
>
> 4. **Training Materials & Annotating Platform**:
>
>     We plan to release the training materials we used to guide the annotators. Meanwhile, the release of the annotating platform will
>     be more difficult. We will try our best to deliver an open-source version of the annotating platform. However, we cannot make any
>     promise at this time.

---

### Official Review · Reviewer_EsaC · 2023-10-30

**Soundness:** 2 fair
**Presentation:** 3 good
**Contribution:** 3 good
**Rating:** 6
**Confidence:** 4

**Summary:**

The paper proposes an evaluation-only benchmark, named MMBench, for multimodal (vision-and-language) models. The benchmark contains 3k single-choice questions for images that come from existing datasets or newly collected sources, covering 20 different ability dimensions. To evaluate vision-language models on the benchmark, the paper proposes an evaluation pipeline featuring the CircularEval strategy, which tests the VLM for multiple times and requires consistent correct answers, and chatGPT-involved choice extraction, which extracts single-choice answers for the VLM responses that do not follow the instruction format well. Multiple models are evaluated on the benchmark, where Qwen-VL-Chat shows the best performance.

**Strengths:**

1. The paper comes with a relatively big (3k) and well-designed benchmark for VLM evaluation, which is an important contribution.
2. Evaluation strategies are designed to test the VLMs that cannot generate single-choice answers. ChatGPT is used in this case, with an analysis compared to human evaluation to show that the introduction of ChatGPT does lead to evaluation bias.
3. The paper is well-written and easy to follow.

**Weaknesses:**

1. More discussions of the 20 different ability dimensions would be favored. How these dimensions are selected can be discussed further. Moreover, in many cases, multiple abilities are entangled with each other in order to correctly answer a question. For example, “How many apples are there in the image?” as shown in Fig-3 requires both numerical (counting) reasoning and perception (detect apples), which category does this example belong to?
2. The results are usually “winner takes all”. As shown in the results in Tab-3, more powerful models are usually stronger in every evaluation dimension. It would be interesting to see more fine-grained analysis, e.g. some models are stronger in dimension A, while another is stronger in dimension B, etc.
3. Bias analysis. Shortcut/bias has long been a problem in VQA, where language bias and visual context bias are entangled with each other, leading the models to take shortcuts to answer the questions without real understanding. Does this benchmark suffer from similar problems? Some analysis on how the dataset is balanced, as well as visualizations of the distribution for different concepts and how they co-occur with each other would be helpful.
4. It would be good to have the results for Bard and GPT-4V.

**Questions:**

See weakness.

**Details Of Ethics Concerns:**

A new dataset is introduced.

---

> ### Author Response · Authors · 2023-11-17
>
> Thank you for your valuable advice. Here is our response to your questions:
>
> ## Q1: How these existing 20 ability dimensions are selected
>
> 1. Perception and reasoning are two crucial cognitive abilities in humans. Firstly, we refer to the definitions of perception and reasoning in biology, such as the specific aspects they encompass. The part has been thoroughly discussed in Section 2.1 of the main paper.
>
> 2. There is currently no consensus in the academic community regarding the specific taxonomy of sub-abilities in perception and reasoning, and there also lacks a comprehensive theoretical framework. Therefore, we gathered a group of experts from the fields of computer vision, natural language processing, and machine learning to discuss the specific ability taxonomy in perception and reasoning. As a result, we obtained 20 ability dimensions included in MMBench, as well as the ability hierarchy ranging from L1 to L3.
>
> 3. The existing 20 ability dimensions are not fixed, and we will continuously update the existing ability dimensions to further improve  the ability taxonomy.
>
> ## Q2: How to classify a question, if multiple ability dimensions are entangled
>
> According to the definition in Page 16, we categorize the "counting" ability as "Object Localization". In MMBench, we do not consider "counting" as a more complex reasoning problem, since for most counting questions, there only exist several objects to be counted.
>
> If multiple ability dimensions are intertwined, we consistently categorize the current question under the more challenging ability dimension. This is because, in most scenarios, the more difficult ability dimension necessitates the model to possess the capability to solve simpler tasks. By doing so, it can effectively utilize the task outputs as intermediate results to tackle more complex tasks. For instance, all questions in "Attribute Comparison" rely on the capability of "Attribute Recognition", since we consider "Attribute Comparison" as a more complex task, we categorize these questions as "Attribute Comparison".
>
> ## Q3: The results are usually “winner takes all”
> Table 3 presents the results of the L-2 ability dimension. Unlike a "winner takes all" scenario, these results on L-3 ability dimension offer a nuanced understanding of the model's capabilities. For further insights, please refer to Tables 9 through 14, which provide additional details on the results. For instance, Shikra demonstrates exceptional performance in Action Recognition, while LLaMA-Adapter excels in Attribute comparison. We hypothesize that factors such as data composition, instruction data quality, and instruction template significantly influence a model's performance across different ability dimensions.
>
> ## Q4. Bias Analysis
>
> | Model | Overall | CP | FP-S  | FP-C  | AR | LR | RR  |
> | --- | --- | --- | -- | -- | -- | -- | -- |
> | GPT-4v | 75.06   | 82.77 | 71.65 | 66.43 | 77.88 | 69.29 | 74.31 |
> | Qwen-VL-Chat | 60.57   | 79.39 | 66.21 | 48.25 | 59.8  | 32.2  | 43.48 |
> | Shikra  | 59.36 | 76.35 | 57.68 | 58.74 | 57.29 | 26.27 | 58.26 |
> | Bard  | 58.16 | 58.42 | 57.6  | 39.6  | 68.27 | 53.4  | 71.64 |
> | IDEFICS-80B | 54.81   | 64.53 | 58.36 | 44.76 | 65.83 | 23.73 | 46.09 |
> | GPT-4  | 13.23 | 6.41  | 7.84  | 6.99  | 32.16 | 27.96 | 4.34  |
> | GPT-3.5 | 11.17 | 2.7   | 9.9   | 4.9   | 22.11 | 25.42 | 10.43 |
> | Claude-2 | 10.74 | 5.74  | 6.14  | 2.8   | 27.14 | 22.88 | 4.35  |
>
> One shortcut might be that the model could provide accurate answers even without relying on image information. To assess the necessity of image data for questions in our dataset, we employed some state-of-the-art LLMs to respond to queries in the MMBench-dev. During the evaluation, we design meta prompts to guide the LLMs try their best to make a reasonable guess, even if a question is not theoretically answerable given only the text part.
>
> As demonstrated in above table, the performance of the text-only GPT-4 model — prompted solely with question text — is notably suboptimal. This limitation is particularly evident in responding to queries that require 'perception' abilities. Such questions, often intrinsically linked to visual information, include examples like "What mood does this image convey?" or "Who is this person?" In contrast, GPT-4 exhibits a degree of proficiency in 'reasoning' tasks, with a notable strength in logical reasoning (LR). The model's performance in LR is comparable to that of other leading open-source models. The effectiveness in LR could be attributed to the model's ability to leverage common sense in answering questions. Notably, some LR data, sourced from ScienceQA, suggest that a portion of these tasks can be effectively addressed using the model's inherent common sense reasoning [1]. In future work, we will meticulously curate our dataset to guarantee that the image data in MMBench is pivotal for addressing every question.
>
> [1] Learn to Explain: Multimodal Reasoning via Thought Chains for Science Question Answering

---

> > ### Author Response · Authors · 2023-11-17
> >
> > ## Q4. Bias Analysis (cont.)
> >
> > | Model            | Dev   | Dev w/o. GPT-4 | Test  | Test w/o. GPT-4 |
> > | ---------------- | ----- | -------------- | ----- | --------------- |
> > | Qwen-VL-Chat     | 60.57 | 59.9           | 61.83 | 60.75           |
> > | Shikra           | 59.36 | 60             | 60.43 | 60.81           |
> > | IDEFICS-80B      | 54.81 | 51.29          | 54.82 | 52.34           |
> > | IDEFICS-9B       | 48.37 | 47.03          | 45.52 | 44.8            |
> > | mPLUG-Owl        | 48.11 | 48.51          | 46.41 | 46.11           |
> > | LLaVA            | 44.5  | 42.77          | 42.21 | 41.74           |
> > | InstructBLIP-13B | 44.42 | 43.17          | 43.39 | 43.18           |
> > | MiniGPT-4-13B    | 42.53 | 40.5           | 42.54 | 41.62           |
> > | LLaMA-Adapter    | 41.24 | 41.09          | 39.63 | 39.63           |
> > | VisualGLM        | 38.57 | 37.82          | 33.63 | 33.52           |
> > | Qwen-VL          | 38.23 | 36.53          | 32.23 | 31.34           |
> > | InstructBLIP     | 35.48 | 33.66          | 35.37 | 34.08           |
> > | PandaGPT         | 33.93 | 32.77          | 30.72 | 29.1            |
> > | MiniGPT-4        | 32.04 | 30.69          | 29.43 | 28.04           |
> > | MMGPT            | 15.21 | 13.66          | 15.98 | 15.26           |
> > | OpenFlamingo v2  | 6.7   | 5.45           | 5.72  | 4.61            |
> > | OpenFlamingo     | 4.64  | 3.37           | 4.54  | 3.55            |
> >
> > Meanwhile, we have also removed the questions that were correctly answered by GPT-4 and re-calculate the accuracies. The results are demonstrated in the table below. Basically, removing those questions has limited impact on the final accuracy, and will not change the conclusions drawn in this work.
> >
> > ## Q5. Results for closed source VLMs (GPT-4v, Bard, etc.)
> >
> > Currently, we have evaluated GPT-4v and Bard on MMBench-Dev, and we will provide the results on MMBench-Test in the final version. Note that due to the strategies set by developers, the closed source VLMs may reject some of the questions: GPT-4v refused to answer 4.2% questions in MMBench-Dev, Bard refused to answer 31.6% questions in MMBench-Dev. In Table 1, we report the accuracies of GPT-4v and Bard (only answered questions count: Acc = (Number of questions answered correctly) / (Number of questions answered)).
> >
> > | Model    | Overall | CP    | FP-S  | FP-C  | AR    | LR    | RR    |
> > | ------------ | ------- | ----- | ----- | ----- | ----- | ----- | ----- |
> > | GPT-4v\*     | 75.06   | 82.77 | 71.65 | 66.43 | 77.88 | 69.29 | 74.31 |
> > | Qwen-VL-Chat | 60.57   | 79.39 | 66.21 | 48.25 | 59.8  | 32.2  | 43.48 |
> > | Shikra       | 59.36   | 76.35 | 57.68 | 58.74 | 57.29 | 26.27 | 58.26 |
> > | Bard\*       | 58.16   | 58.42 | 57.6  | 39.6  | 68.27 | 53.4  | 71.64 |
> > | IDEFICS-80B  | 54.81   | 64.53 | 58.36 | 44.76 | 65.83 | 23.73 | 46.09 |
> >
> > **Table 1. Model accuracies on MMBench-Dev (all questions). * mean the question numbers used for calculating accuracies may be different from other models.**
> >
> > In Table 2, 3, we provide apple-to-apple accuracy comparisons for GPT-4v and Bard, respectively. In those comparisons, the advantages of GPT-4v and Bard are more significant.
> >
> > | Model    | Overall | CP    | FP-S  | FP-C  | AR    | LR    | RR    |
> > | ------------ | ------- | ----- | ----- | ----- | ----- | ----- | ----- |
> > | GPT-4v\*     | 75.06   | 82.77 | 71.65 | 66.43 | 77.88 | 69.29 | 74.31 |
> > | Qwen-VL-Chat | 59.46   | 79.39 | 61.42 | 48.25 | 59.8  | 33.33 | 42.2  |
> > | Shikra       | 59.1    | 76.35 | 56.3  | 58.74 | 57.29 | 26.32 | 56.88 |
> > | IDEFICS-80B  | 53.81   | 64.53 | 53.15 | 44.76 | 65.83 | 24.56 | 46.79 |
> >
> > **Table 2. Model accuracies on MMBench-Dev (only questions answered by GPT-4v count).**
> >
> > | Model    | Overall | CP    | FP-S  | FP-C  | AR    | LR    | RR    |
> > | ------------ | ------- | ----- | ----- | ----- | ----- | ----- | ----- |
> > | Bard\*       | 58.16   | 58.42 | 57.6  | 39.6  | 68.27 | 53.4  | 71.64 |
> > | Qwen-VL-Chat | 56.28   | 80.9  | 62.67 | 36.63 | 53.1  | 34.09 | 35.82 |
> > | Shikra       | 52.89   | 74.72 | 57.14 | 48.51 | 48.28 | 20.45 | 40.3  |
> > | IDEFICS-80B  | 49.62   | 65.73 | 54.38 | 28.71 | 59.31 | 22.73 | 37.31 |
> >
> > **Table 3. Model accuracies on MMBench-Dev (only questions answered by Bard count).**

---

> > > ### Author Response · Authors · 2023-11-20
> > > **Looking forward to your reply!**
> > >
> > > We sincerely appreciate your great efforts in reviewing this paper. Your constructive advice and valuable comments really help improve our paper. Considering the approaching deadline, please, let us know if you have follow-up concerns. We sincerely hope you can consider our reply in your assessment, and we can further address unclear explanations and remaining concerns if any.
> > >
> > > Once more, we are appreciated for the time and effort you've dedicated to our paper.

---

> > > > ### Author Response · Authors · 2023-11-21
> > > > **Sincere Request for Further Discussions**
> > > >
> > > > Thanks again for your great efforts in reviewing this paper! With the discussion period drawing to a close, we expect your feedback and thoughts on our reply. We put a significant effort into our response and sincerely hope you can consider our reply in your assessment. We look forward to hearing from you, and we can further address unclear explanations and remaining concerns if any.

---

> ### Comment · Reviewer_EsaC · 2023-11-22
> **Thanks for the response**
>
> I thank the authors for the response. I appreciate the GPT-4V results, as well as the results for the bias.
>
> The text-bias result is interesting. In bias results, it's shown that text-only GPT-4 without looking at the images gets 13% accuracy, with a 28% LR (reasoning) performance that is higher than some models with images. It will be interesting to see similar analysis for more models besides GPT-4.
>
> > In Table 1, we report the accuracies of GPT-4v and Bard (only answered questions count: Acc = (Number of questions answered correctly) / (Number of questions answered)).
>
> I believe this metric is misleading and leads to misleading high results for models that produces "not answerable" answers for most of the questions. A better metric is to assign random answers for the not answerable questions, then calculate accuracy. For example, Bard refused to answer 31.6% questions, which is not a small percentage.
>
> Additionally, analysis of those rejected questions, i.e. whether they are ambiguous, low-quality, or some other reasons, would be favorable.
>
> Moreover, the response does not address my concerns in a more detailed analysis on analysis of the 20 different ability axis. As most of the results demonstrate "winner takes all". Since the benchmark introduces the 20 dimensions, showing the analysis findings on the 20 dimensions is necessary to show the advantage of reporting 20 numbers over only the overall accuracy.
>
> > please refer to Tables 9 through 14, which provide additional details on the results. For instance, Shikra demonstrates exceptional performance in Action Recognition, while LLaMA-Adapter excels in Attribute comparison. We hypothesize that factors such as data composition, instruction data quality, and instruction template significantly influence a model's performance across different ability dimensions.
>
> While the authors provide intensive results in Tab 9-14, the results are less meaningful if no analysis is provided. The analysis provided here is too vague and general to show the necessity for evaluating on different dimensions.
>
> Overall, I thank the authors for their efforts in the rebuttal and the new results. With the remaining concerns, I keep my rating unchanged.

---

> > ### Author Response · Authors · 2023-11-23
> > **Some quick facts on the rejected questions.**
> >
> > Dear reviewer, we would like to supplement some quick facts on the rejected questions.
> > We have browsed the rejected questions by Bard and GPT-4v. Actually, we believe that most rejected questions are rejected due to the policy set by the provider organization, and it doesn't mean that Bard or GPT-4v are not capable to solve the questions.
> > For example, we find that Bard will reject a large proportion of questions with images that contain recognizable human beings, while GPT-4v will reject many questions with images including celebrities. Comparing Bard or GPT-4v with other VLMs can be challenging under such situation, and we think the provided Table 2, 3 in the rebuttal can present a reasonable comparison setting.

---

### Official Review · Reviewer_Ntup · 2023-10-30

**Soundness:** 3 good
**Presentation:** 3 good
**Contribution:** 3 good
**Rating:** 6
**Confidence:** 3

**Summary:**

This paper aims to establish a new benchmark, known as MMBench, for evaluating the multi-modal capabilities of VLMs. In comparison to previous benchmarks, MMBench offers a fine-grained assessment of abilities and employs more robust evaluation metrics. This is achieved by incorporating a wider range of evaluation questions. Additionally, MMBench introduces a rigorous CircularEval strategy that ensures models comprehend the questions and provide answers based on understanding rather than guessing. Moreover, MMBench leverages ChatGPT to convert open-form predictions into pre-defined options, mitigating the impact of varying instruction-following capabilities of VLMs. The proposed benchmark evaluates various MLLMs, revealing their capabilities and limitations.

**Strengths:**

The current MLLMs greatly require a fair and reasonable benchmark to assess the strengths and weaknesses of different methods, making the problem addressed in this paper highly significant. The proposed CircularEVAL strategy effectively enhances the robustness of the evaluations.

**Weaknesses:**

The authors should provide results for GPT4 to establish the upper bound of performance within the proposed benchmark.

For tasks that perform poorly within the current benchmark, the authors should explain why the models exhibit such poor performance. Is it due to inherent issues with the tasks themselves? Additionally, a comparison with the results of GPT4 can be made to analyze the performance shortcomings of the current open-source MLLMs.

**Questions:**

No other questions.

---

> ### Author Response · Authors · 2023-11-17
>
> Thank you for your valuable advice. Here is our response to your questions:
>
> ## Q1: Results for closed source VLMs (GPT-4v, Bard, etc.)
>
> Currently, we have evaluated GPT-4v and Bard on MMBench-Dev, and we will provide the results on MMBench-Test in the final version. Note that due to the strategies set by developers, the closed source VLMs may reject some of the questions: GPT-4v refused to answer 4.2% questions in MMBench-Dev, Bard refused to answer 31.6% questions in MMBench-Dev. In Table 1, we report the accuracies of GPT-4v and Bard (only answered questions count: Acc = (Number of questions answered correctly) / (Number of questions answered)).
>
> | Model    | Overall | CP    | FP-S  | FP-C  | AR    | LR    | RR    |
> | ------------ | ------- | ----- | ----- | ----- | ----- | ----- | ----- |
> | GPT-4v\*     | 75.06   | 82.77 | 71.65 | 66.43 | 77.88 | 69.29 | 74.31 |
> | Qwen-VL-Chat | 60.57   | 79.39 | 66.21 | 48.25 | 59.8  | 32.2  | 43.48 |
> | Shikra       | 59.36   | 76.35 | 57.68 | 58.74 | 57.29 | 26.27 | 58.26 |
> | Bard\*       | 58.16   | 58.42 | 57.6  | 39.6  | 68.27 | 53.4  | 71.64 |
> | IDEFICS-80B  | 54.81   | 64.53 | 58.36 | 44.76 | 65.83 | 23.73 | 46.09 |
>
> **Table 1. Model accuracies on MMBench-Dev (all questions). * mean the question numbers used for calculating accuracies may be different from other models.**
>
> In Table 2, 3, we provide apple-to-apple accuracy comparisons for GPT-4v and Bard, respectively. In those comparisons, the advantages of GPT-4v and Bard are more significant.
>
> | Model    | Overall | CP    | FP-S  | FP-C  | AR    | LR    | RR    |
> | ------------ | ------- | ----- | ----- | ----- | ----- | ----- | ----- |
> | GPT-4v\*     | 75.06   | 82.77 | 71.65 | 66.43 | 77.88 | 69.29 | 74.31 |
> | Qwen-VL-Chat | 59.46   | 79.39 | 61.42 | 48.25 | 59.8  | 33.33 | 42.2  |
> | Shikra       | 59.1    | 76.35 | 56.3  | 58.74 | 57.29 | 26.32 | 56.88 |
> | IDEFICS-80B  | 53.81   | 64.53 | 53.15 | 44.76 | 65.83 | 24.56 | 46.79 |
>
> **Table 2. Model accuracies on MMBench-Dev (only questions answered by GPT-4v count).**
>
> | Model    | Overall | CP    | FP-S  | FP-C  | AR    | LR    | RR    |
> | ------------ | ------- | ----- | ----- | ----- | ----- | ----- | ----- |
> | Bard\*       | 58.16   | 58.42 | 57.6  | 39.6  | 68.27 | 53.4  | 71.64 |
> | Qwen-VL-Chat | 56.28   | 80.9  | 62.67 | 36.63 | 53.1  | 34.09 | 35.82 |
> | Shikra       | 52.89   | 74.72 | 57.14 | 48.51 | 48.28 | 20.45 | 40.3  |
> | IDEFICS-80B  | 49.62   | 65.73 | 54.38 | 28.71 | 59.31 | 22.73 | 37.31 |
>
> **Table 3. Model accuracies on MMBench-Dev (only questions answered by Bard count).**
>
> ## Q2: Performance Analysis & Comparison with closed source VLMs
>
> 1. In Table 1, GPT-4v significantly outperforms the leading open-source model (Qwen-VL-Chat) by over 14% overall accuracy on MMBench-Dev. The improvement is primarily evident in reasoning tasks:
>     1. In Attribute Reasoning questions, GPT-4v outperforms the second best opensource VLM IDEFICS-80B by over 12% Top-1 accuracy.
>     2. In Logic Reasoning questions, GPT-4v outperform1s all opensource VLMs by over 35% Top-1 accuracy
>     3. In Relation Reasoning questions, GPT-4v outperforms all opensource VLMs by over 15% Top-1 accuracy
>      Meanwhile, the gap between opensource VLMs and GPT-4v is not that huge on perceptions tasks (3% - 10% Top-1 accuracy).
> 2. Under apple-to-apple comparisons (only questions answered by the closed source VLM count), the advantage of closed source VLMs is more significant. Generally, the perception performance of opensource VLMs is good and comparable with GPT-4v or Bard (under some preception tasks, opensource VLMs even significantly outperforms Bard). However, the reasoning performance of opensource VLMs still lag far behind GPT-4v or Bard. We attribute the superior reasoning capability of GPT-4v or Bard to their potentially larger and more powerful language models.
> 3. Most existing opensource VLMs simply map the visual concepts to the language embedding space, and finetune the model with VL instruction data (with a large proportion of questions perception related). To further improve the reasoning capabilities, the developers can: 1. Switch to more powerful language backbones; 2. Design better finetuning algorithms; 3. Built reasoning-related instruction datasets with better quality.

---

> > ### Author Response · Authors · 2023-11-20
> > **Looking forward to your reply!**
> >
> > We sincerely appreciate your great efforts in reviewing this paper. Your constructive advice and valuable comments really help improve our paper. Considering the approaching deadline, please, let us know if you have follow-up concerns. We sincerely hope you can consider our reply in your assessment, and we can further address unclear explanations and remaining concerns if any.
> >
> > Once more, we are appreciated for the time and effort you've dedicated to our paper.

---

> > > ### Author Response · Authors · 2023-11-21
> > > **Sincere Request for Further Discussions**
> > >
> > > Thanks again for your great efforts in reviewing this paper! With the discussion period drawing to a close, we expect your feedback and thoughts on our reply. We put a significant effort into our response and sincerely hope you can consider our reply in your assessment. We look forward to hearing from you, and we can further address unclear explanations and remaining concerns if any.

---

### Meta-Review · Area_Chair_YBup · 2023-12-11

**Metareview:**

The paper presents a new multiple-choice VQA evaluation benchmark for assessing recent multimodal language large models without subjective human evaluation. The benchmark is set up to evaluate the perception and reasoning abilities of these models, such as attribute prediction, OCR, action recognition, social relation, and so on.

While the authors effectively addressed some major concerns raised during the rebuttal stage, certain concerns remain. For example,  some reviewers still have concerns about the data quality of the constructed benchmark. Additionally, there's a concern that a new VLM/prompt engineer might deliberately choose to ignore the multiple choices and directly output the final answer, thus leaving all answer extraction to the powerful ChatGPT.

Based on those concerns, we recommend rejecting the current version of submission and strongly recommend authors taking reviewers' suggestions to refine their work.

**Justification For Why Not Higher Score:**

As a benchmark work, several reviewers still have significant concerns regarding the data quality, data source, and effectiveness of the constructed benchmark when encountering intentional manipulation by a VLM/prompt engineer.

**Justification For Why Not Lower Score:**

N/A

---

### Decision · Program_Chairs · 2024-01-16

Reject